# Bayesian Coreset Optimization for Personalized Federated Learning

**Prateek Chanda**
Department of Computer Science
Indian Institute of Technology
Bombay, India
`prateekch@cse.iitb.ac.in`

**Shrey Modi**
Department of Computer Science
Indian Institute of Technology
Bombay, India
`200020135@iitb.ac.in`

**Ganesh Ramakrishnan**
Department of Computer Science
Indian Institute of Technology
Bombay, India
`ganesh@cse.iitb.ac.in`

## Abstract

In a distributed machine learning setting like Federated Learning where there are multiple clients involved which update their individual weights to a single central server, often training on the entire individual client's dataset for each client becomes cumbersome. To address this issue we propose CORESET-PFEDBAYES: a personalized coreset weighted federated learning setup where the training updates for each individual clients are forwarded to the central server based on only individual client coreset based representative data points instead of the entire client data. Through theoretical analysis we present how the average generalization error is minimax optimal up to logarithm bounds (upper bounded by $\mathcal{O}(n_k^{-\frac{2\beta}{2\beta+\Lambda}} \log^{2\delta'}(n_k)))$ and lower bounds of $\mathcal{O}(n_k^{-\frac{2\beta}{2\beta+\Lambda}})$, and how the overall generalization error on the data likelihood differs from a vanilla Federated Learning setup as a closed form function $\mathfrak{F}(\boldsymbol{w}, n_k)$ of the coreset weights $\boldsymbol{w}$ and coreset sample size $n_k$. Our experiments on different benchmark datasets based on a variety of recent personalized federated learning architectures show significant gains as compared to random sampling on the training data followed by federated learning, thereby indicating how intelligently selecting such training samples can help in performance. Additionally, through experiments on medical datasets our proposed method showcases some gains as compared to other submodular optimization based approaches used for subset selection on client's data.

## 1 Introduction

Distributed machine learning has a wide variety of applications in various domains like in recommendation systems (Zhang et al., 2022a), healthcare and other areas. Given the advantages of data-privacy, heterogeneity and resource efficiency, federated learning in particular stands out among all other learning methods. For instance in recommendation systems based work (Luo et al., 2022) (Zhang et al., 2023) or in healthcare (Lu et al., 2022), personalization and privacy preserving based recommendations are currently most focused upon. However, often these require inference on larger datasets which is often computationally expensive given the fact that often these datasets only consist of a subset of representative data points (Kaushal et al., 2018), (Maheshwari et al., 2020).

Recently, researchers have tried to use Bayesian coresets for performing inference (Campbell & Beronov, 2019),(Campbell & Broderick, 2018). Bayesian coreset refers to a subset of the complete dataset which can be used to approximate the posterior inference as if it was performed on the complete dataset. In Federated Learning, we have a distributed machine learning model where there are a set of clients each having their own share of data and a server. This is prevalent in many production level systems such as recommendation systems or machine learning models deployed for mobile

apps. It is imperative for each client to get the same user satisfiability and personalized experience with only a small chunk of data on their local devices. Instead of training on the entire user data for a particular client, what if we can only learn based on a representative set of the user data for each client and achieve near optimal accuracy? In this space (Huang et al., 2022) perform coresets optimization for vertical federated learning where empirically they show that coresets optimization help reducing the communication complexity among clients and server in the vertical FL setting.

Our contributions are as follows:

- Proposal of a new architecture to incorporate bayesian coresets optimization in the space of Federated Learning.
- Proposal of multiple novel objective functions that take into account the optimization problem of general Federated Learning in a Bayesian coresets setting with particular focus on personalized coreset weights for each individual clients.
- Theoretical analysis on the convergence rate shows our approach CORESET-PFEDBAYES achieves convergence rate within logarithmic bounds.
- Experimental results across several benchmark datasets conducted with a wide array of pre-existing baselines show promising results towards good performance in terms of model accuracy even with less data at each client's end.

## 2 RELATED WORK

It is well known in literature that training datasets offer diminishing returns in terms of performance. It has also been demonstrated that one can train and obtain better returns in performance and energy efficiency by training models over subsets of data that are meticulously selected (Ghorbani & Zou, 2019; Yoon et al., 2020; Katharopoulos & Fleuret, 2018; Strubell et al., 2019). This leads us to the problem of coreset selection that deals with approximating a desirable quantity (*e.g.*, gradient of a loss function) over an entire dataset with a weighted subset of it. Traditional methods of coreset selection have used a submodular proxy function to select the coreset and are model dependent (Wei et al., 2015; Kirchhoff & Bilmes, 2014; Kaushal et al., 2019; Har-Peled & Mazumdar, 2004; Mirzasoleiman et al., 2015). Coreset selection with deep learning models has become popular in recent years (Mirzasoleiman et al., 2020a;b; Killamsetty et al., 2021; Coleman et al., 2019; Owen & Daskin, 1998). Coreset selection to approximately match the full-batch training gradient is proposed in (Mirzasoleiman et al., 2020a). Killamsetty et al. (2021) propose algorithms to select coresets that either match the full-batch gradient or the validation gradient. Mirzasoleiman et al. (2020b) propose an approach to select a coreset that admits a low-rank jacobian matrix and show that such an approach is robust to label noise. Most existing coreset selection approaches are proposed in conventional settings wherein all the data is available in one place. In FL, since no client or the server gets a holistic view of the training dataset, coresets can at best approximate only local data characteristics and are thus inherently sub-optimal.

Coreset selection in federated learning has remained largely unexplored because of the intricacies involved due to privacy and data partition across clients. Federated Learning can be modelled as a cooperative game where it often uses Shapley values to select clients whose updates result in the best reduction of the loss on a validation dataset held by the server. One work that comes very close to ours is that of (Balakrishnan et al., 2021), that selects a coreset of clients[1] whose update represents the update aggregated across all the clients. They apply facility location algorithm on the gradient space to select such a coreset of clients. In contrast to all these approaches, our proposal attempts to select a coreset of the dataset at each client and is thus more fine-grained than such prior works. One another paradigm of FL that is in contrast to our setup is Personalized FL, where the aim is to train specialised models for each individual client (Collins et al., 2021; Fallah et al., 2020a; Marfoq et al., 2022; Li et al., 2022; Jain et al., 2021). While personalized FL focuses on finetuning the model to match each client's data distribution, we build models to account for just the server's distribution, similar to (Karimireddy et al., 2020).

The algorithms that ensure privacy in FL include differential privacy (Dwork et al., 2006; Kairouz et al., 2021), Homomorphic encryption (Segal et al., 2017; Li et al., 2020), *etc.*; which is not the

---

[1]As against selecting a coreset of data instances

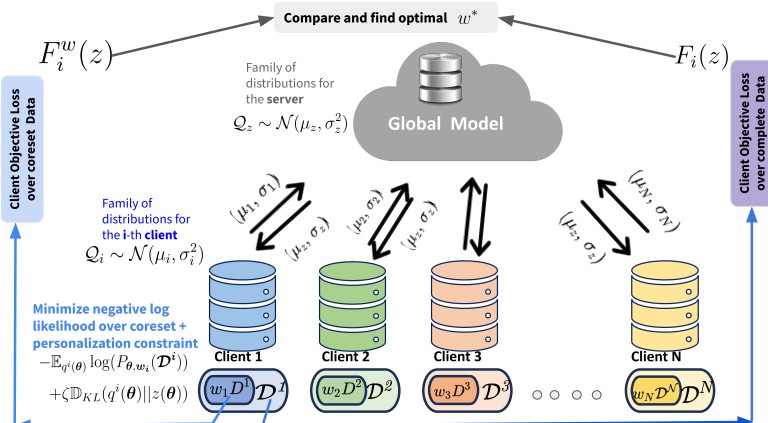

Figure 1: **System Diagram** : Coreset Weighted Personalized Federated Learning model with parameters under Gaussian assumptions. Each client uploads its updated distribution to the server based its corresponding coreset training data (each client $i$'s data $\mathcal{D}^i$ is weighted by $\boldsymbol{w}_i$) and then the aggregated global distribution is utilised from the server.

focus of our work. However, we note that all these methods can be easily integrated with our solution approach and can be used in conjunction.

## 3 PRELIMINARIES AND PROBLEM SETTING

We consider the problem setting from (Zhang et al., 2022b) as follows: Consider a distributed system that includes one server and $N$ clients. Let the $i$th client's dataset be $\mathcal{D}^i = \{(\boldsymbol{x}_j^i, \boldsymbol{y}_j^i)\}_{j=1}^n$.(assuming all $N$ clients have the same sample size $n$) Further, let the $i$-th client satisfy a regression model with random covariates as $\boldsymbol{y}_j^i = f^i(\boldsymbol{x}_j^i) + \epsilon_j^i, \quad \forall j \in [1, \dots n]$ where $\boldsymbol{x}_j^i \in \mathbb{R}^{s_0}, \boldsymbol{y}_j^i \in \mathbb{R}^{s_L+1}$ for $\forall i = 1, \dots, N$. Here $f^i(\bullet) : \mathbb{R}^{s_0} \to \mathbb{R}^{s_L+1}$ denotes a nonlinear function (which is unknown and we want to estimate) and $\epsilon_j^i$ denotes a Gaussian noise independent of $\boldsymbol{x}_j^i$ i.e. $\epsilon_j^i \overset{\text{i.i.d}}{\sim} \mathcal{N}(0, \sigma_\epsilon^2)$ with variance $\sigma_\epsilon^2$. We assume that $f^i(\bullet)$ is $\beta$-Hölder-smooth functions (Takezawa, 2005) and the intrinsic dimension of each of the client's data is $\boldsymbol{\Lambda}$. We assume each client has the same fully-connected deep neural network (DNN), each having their individual neural network parameters. We denote the output of the neural network as $f_{\boldsymbol{\theta}}(\bullet)$ where $\boldsymbol{\theta} \in \mathbb{R}^T$ represents the network parameters. Analogously the output of the $i$-th client is denoted as $f_{\boldsymbol{\theta}}^i$. Each neural network has $L$ hidden layers, where the $j$-th hidden layer has $s_j$ neurons and its corresponding activation functions $\sigma(\bullet)$. We denote $\boldsymbol{\mathcal{S}} = \{s_i\}_{i=1}^L$, the set of all neurons in the neural network. The assumption made is that all the neural network parameters are bounded *i.e.*, $||\boldsymbol{\theta}||_\infty \leq \Omega$ for $\Omega > 0$.

The main aim is to architect a Federated Learning system that takes into consideration the coreset problem for each individual client along with personalization and tackling overfitting. Using the above problem setting we first formulate the client side objective and server side objective for the Federated Learning setting as a bi-level optimization problem.

### 3.1 FEDERATED LEARNING OBJECTIVES

The standard BNN model Jordan et al. (1999) aims to solve the optimization problem to find the closest distribution $q^i(\boldsymbol{\theta})$ for the $i$-th client from the family of distributions $\mathcal{Q}_i$ to match the posterior distribution $\pi(\boldsymbol{\theta}|\mathcal{D}^{\boldsymbol{i}})$ of the given data $\mathcal{D}^{\boldsymbol{i}}$ via minimizing the KL-divergence as follows.

**Client Side Objective**

$$\mathcal{F}_i(z) \triangleq \min_{q^i(\boldsymbol{\theta}) \in \mathcal{Q}_i} \mathbb{D}_{KL}(q^i(\boldsymbol{\theta})||\pi(\boldsymbol{\theta}|\mathcal{D}^i)) \Leftrightarrow \min_{q^i(\boldsymbol{\theta}) \in \mathcal{Q}_i} \overbrace{-\mathbb{E}_{q^i(\boldsymbol{\theta})}[\log P_{\boldsymbol{\theta}}(\mathcal{D}^i)]}^{\text{reconstruction error over } \mathcal{D}} + \zeta \overbrace{\mathbb{D}_{KL}(q^i(\boldsymbol{\theta})||\pi(\boldsymbol{\theta}))}^{\text{regularization term}}$$

$$(1)$$

Here $\pi(\boldsymbol{\theta})$ denotes the prior distribution and $P_{\boldsymbol{\theta}}(\mathcal{D}^i)$ denotes the likelihood and $\zeta$ is a personalization constant that defines the weightage towards more regularization thus leading to more personalization.

**Server Side Objective** On the server side the global model tries to find the closest distribution in $\mathcal{Q}_z$ to the client's distribution by minimizing the aggregate KL divergence from all the clients as follows:

$$\min_{z(\boldsymbol{\theta}) \sim \mathcal{Q}_z} \mathcal{F}(z) \triangleq \frac{1}{N} \sum_{i=1} \mathcal{F}_i(z) \tag{2}$$

### 3.2 BAYESIAN CORESETS OPTIMIZATION

We now introduce the notion of coreset weights i.e. we assign to each client $i$'s data $\mathcal{D}^i$ a weight vector $\boldsymbol{w}_i \in \mathbb{R}^n$ that will act as the corresponding coreset weight for the $i$-th client. In standard bayesian coresets optimization setting, the goal is to control the deviation of coreset log-likelihood from the true log-likelihood via some sparsity ($n_k << n$). In accordance to (Zhang et al., 2021) we utilise the following optimization objective for the $i$-th client:

$$\arg \min_{\boldsymbol{w}_i \in \mathbb{R}^n} \mathcal{G}^i(\boldsymbol{w}_i) := \left\| \mathcal{P}_{\boldsymbol{\theta}}(\mathcal{D}^i) - \mathcal{P}_{\boldsymbol{\theta}, \boldsymbol{w}_i}(\mathcal{D}^i) \right\|_{\hat{\pi}, 2}^2 \quad s.t. \quad ||\boldsymbol{w}_i||_0 \leq n_k, \quad \forall i \in [N] \tag{3}$$

where the coreset weights $\boldsymbol{w}_i$ are considered over the data points $\mathcal{D}^i$ and $L^2(\hat{\pi})$-norm as the distance metric is considered in the embedding Hilbert Space. Specifically, $\hat{\pi}$ is the weighting distribution that has the same support as true posterior $\pi$. The above equation can be further approximated to the following where $\hat{g}_j$ is a Monte-Carlo approximation over $g_j = \mathcal{P}_{\boldsymbol{\theta}}(\mathcal{D}_j^i) - \mathbb{E}_{\theta \sim \hat{\pi}} P_\theta(\mathcal{D}_j^i)$ for Monte-carlo samples (the derivation can be found in Appendix 2)

$$\arg \min_{\boldsymbol{w} \in \mathbb{R}^n} \mathcal{G}^i(\boldsymbol{w}_i) := \left\| \sum_{i=1}^n \hat{g}_j - \sum_{i=1}^n \boldsymbol{w}_i \hat{g}_j \right\|_2^2 \quad s.t. \quad ||\boldsymbol{w}_i||_0 \leq n_k, \quad \forall i \in [N] \tag{4}$$

The above *sparse regression* problem is non-convex due to the combinatorial nature of the constraints. (Campbell & Broderick, 2018) uses the $l_2$-norm formulation which however results in less approximation accuracy compared to (Campbell & Beronov, 2019). As the authors in (Zhang et al., 2021) point out both the above approaches have expensive computation cost and hence they propose a better alternative via accelerated iterative thresholding Appendix 10.7.

## 4 METHODOLOGY - CORESET-PFEDBAYES

We now proceed towards formulating our problem for combining the coreset optimization problem in a federated learning setting.

### 4.1 MODIFIED FEDERATED LEARNING OBJECTIVES - INCORPORATING CORESET WEIGHTS

**Modified Client Side Objective** We now aim towards incorporating the coreset formulation from Eq: 3 in our federated learning setting Eq: 1. Assuming the bayesian coreset weights setup for each client $i$ from , we introduce a new modified client objective function

$$\mathcal{F}_i^w(z) \triangleq \min_{q^i(\boldsymbol{\theta}) \sim \mathcal{Q}_i} [-\mathbb{E}_{q^i(\boldsymbol{\theta})} \log(P_{\boldsymbol{\theta}, \boldsymbol{w}_i}(\mathcal{D}^i)) + \zeta \mathbb{D}_{KL}(q^i(\boldsymbol{\theta})||z(\boldsymbol{\theta}))] \tag{5}$$

where $z(\boldsymbol{\theta})$ and $q^i(\boldsymbol{\theta})$ denote the global distribution and the local distribution for the $i$-the client that is to be optimized respectively. Here, $\mathcal{F}_i^w(\bullet)$ and $\mathcal{F}_i(\bullet)$ indicates the coreset weighted $i$-th client objective and full data based client objective respectively.

In particular, let the family of distributions of the $i$-th client $\mathcal{Q}_i$ and server $\mathcal{Q}_z$ satisfy

$$\mathcal{Q}_{i,k} \sim \mathcal{N}(\mu_{i,k}, \sigma_{i,k}^2), \quad \mathcal{Q}_{z,k} \sim \mathcal{N}(\mu_{z,k}, \sigma_{z,k}^2) \qquad k = 1, \dots, T \tag{6}$$

where the above gaussian parameters correspond to the mean and variance for the $k$-th parameter of the $i$-th client respectively. Similar holds true for the server. This is a valid assumption commonly used in literature (Blundell et al., 2015) that let's us simplify the evaluation of $KL$ divergence score $\mathbb{D}_{KL}$ between $q^i(\boldsymbol{\theta})$ and $z(\boldsymbol{\theta})$, resulting in a closed form solution.

### 4.2 Novel Objective : Achieving near-optimal client distribution based on optimal coreset weights

Let $z^*(\theta)$ be the optimal variational solution of the problem in Eq: (2) and $\hat{q}^i(\theta)$ be its corresponding variational solution for the $i$'th client's objective in Eq: (1). Now, for the same $z^*(\theta)$, let $\hat{q}^i(\theta; w)$ denote the corresponding variational solution for the weighted coreset client objective in Eq: 5. We want to ensure that the optimal distribution $\hat{q}^i(\theta; w)$ which minimizes the coreset objective does not deviate too much from the original distribution $\hat{q}^i(\theta)$ (which acts a solution for Eq: 1) and hence we want to fixate on $w$ accordingly. This intuition comes from the fact that we want to choose $w$ in such a way that the performance of our federated learning system in terms of accuracy does not drop too further in the face of training on a small percentage of data dictated by the coreset weights $w$ (*in our problem setting we denote coresets weights as $w_i$ for each client $i$*). From here forward, let $\mathcal{F}_i^w(\bullet)_{\mathrm{arg}}$ denote $\hat{q}^i(\theta; w)$ and $\mathcal{F}_i(\bullet)_{\mathrm{arg}}$ denote $\hat{q}^i(\theta)$.

### 4.3 Observations and Motivation for a New Objective

If we observe the client equation Eq: 1 closely, we will see that for each client $i$ we are first fixing $z(\theta)$ in the equation. For a fixed $z(\theta)$, we now search for the optimal distribution $\hat{q}^i(\theta)$ among the whole family of distributions local to that client $\mathcal{Q}^i$. This is similar also for Eq: 5 for the coreset weighted client objective.

Our objective is the learned optimal local distribution for the general client optimization objective should be as close as possible to that of the weighted coreset based client optimization objective for the same value of $z$.

So we want to formulate a new objective function such that for each client $i$ we minimize the divergence between the two optimal distributions resulting from the coreset and normal objective functions.

$$\{w_i^*\} \triangleq \arg\min_w \mathbb{D}_{KL}(\mathcal{F}_i^w(z)_{\mathrm{arg}}||\mathcal{F}_i(z)_{\mathrm{arg}}) \;\; \Leftrightarrow \;\; \arg\min_w \mathbb{D}_{KL}(\hat{q}^i(\theta, w)||\hat{q}^i(\theta)) \quad \|w_i\|_0 \leq n_k \tag{7}$$

### 4.4 Incorporating likelihood into Modified Client Objective

Although the above minimization objective Eq: 7 captures the intuition behind matching the near-optimal performance (accuracy) with only a small coreset of the original client's data to that by using the entire data, this approach does not take into account how close the likelihood of each of the client's coreset weighted data subset $P_{\theta, w_i}(\mathcal{D}^i)$ is to that of the original client's data $P_\theta(\mathcal{D}^i)$.

More specifically, we now want to choose the optimal coreset weights ($w_i$ (personal coreset weights) by not only minimizing the KL divergence between the corresponding client distributions ( $\hat{q}^i(\theta; w)$ and $\hat{q}^i(\theta)$) but also taking into account the closeness of the coreset weighted data likelihood to that of the original likelihood.

$$\{w_i^*\} \triangleq \arg\min_w \mathbb{D}_{KL}(\hat{q}^i(\theta, w)||\hat{q}^i(\theta)) + \left\|P_\theta(\mathcal{D}^i) - P_{\theta, w_i}(\mathcal{D}^i)\right\|_{\hat{\pi}, 2}^2 \quad \|w_i\|_0 \leq n_k \tag{8}$$

## 5 Algorithm

We showcase our complete algorithm for CORESET-PFEDBAYES in Table 1 below In line with (Zhang et al., 2022b) we utilise a reparameterization trick for $\theta$ via variables $\mu$ and $\rho$ i.e. $\theta = h(v, g)$, where $\theta_m = h(v_m, g_m) = \mu_m + \log(1 + \exp(\rho_m)) \cdot g_m$, $g_m \sim \mathcal{N}(0, 1)$, where $m \in [1, \ldots, T]$. For the first term in Equation 5, we use a minibatch stochastic gradient descent to get an estimate for the $i$-th client as follows:

$$\Omega^i(v_w) \approx -\frac{n}{b}\frac{1}{K}\sum_{j=1}^b \sum_{k=1}^K \log p_{h(v_w, g_k)}^i\left(D_j^i\right) + \zeta \mathbb{D}_{KL}\left(q_{v_w}^i(\theta; w)||z_{v_w}(\theta)\right) \tag{9}$$

**Algorithm 1**: CORESET-PFEDBAYES

| Server side Objective & Coreset Update | Client Side Objective |
|---|---|
| **Cloud server executes**: | **ClientUpdate** $(i, w_i, \boldsymbol{v}^t)$ : |
|   **Input** $T, R, S, \lambda, \eta, \beta, b, \boldsymbol{v}^0 = (\boldsymbol{\mu}^0, \boldsymbol{\sigma}^0)$ |   $\boldsymbol{v}_{z,0}^t = v^t$ |
|   **for** $t = 0, 1, \ldots, T-1$ **do** |   **for** $r = 0, 1, \ldots, R-1$ **do** |
|     **for** $i = 1, 2, \ldots, N$ **in parallel do** |     $\boldsymbol{D}_\Lambda^i \leftarrow$ sample a minibatch $\Lambda$ with size $b$ from $\boldsymbol{D}^i$ |
|       $\boldsymbol{v}_i^{t+1} \leftarrow$ **CoresetOptUpdate** $(i, \boldsymbol{v}^t)$ |     $\boldsymbol{D}_{\Lambda,w}^i \leftarrow$ sample a minibatch $\Lambda$ with size $b$ from $w_i \boldsymbol{D}^i$ |
|     $\mathbb{S}^t \leftarrow$ Random subset of clients with size $S$ |     $\boldsymbol{g}_{i,r} \leftarrow$ Randomly draw $K$ samples from $\mathcal{N}(0,1)$ |
|     $\boldsymbol{v}^{t+1} = (1-\beta)\boldsymbol{v}^t + \frac{\beta}{S} \sum_{i \in S^t} \boldsymbol{v}_i^{t+1}$ |     $\Omega^i(\boldsymbol{v}_r^t) \leftarrow$ Use Eq:9 with $\boldsymbol{g}_{i,r}$, $\boldsymbol{D}_\Lambda^i$ and $\boldsymbol{v}_r^t$ |
| **CoresetOptUpdate**$(i, \boldsymbol{v}^t)$ : |     $\nabla_v \Omega^i(\boldsymbol{v}_r^t) \leftarrow$ Back propagation w.r.t $\boldsymbol{v}_r^t$ |
|   $y = P_\theta(D^i), \Phi w^i = P_{\theta, w^i}(D^i)$ |     $\boldsymbol{v}_r^t \leftarrow$ Update with $\nabla_v \Omega^i(\boldsymbol{v}_r^t)$ using GD algorithms |
|   **Objective** $f(\boldsymbol{w}) = \mathbb{D}_{KL}(q_w^i \| q^i) + \|y - \Phi w^i\|_2^2$ |     $\Omega_z^i(\boldsymbol{v}_{z,r}^t) \leftarrow$ Forward propagation w.r.t $\boldsymbol{v}$ |
|   $t = 0, l_0 = 0, w_0^i = 0$ |     $\nabla \Omega_z^i(\boldsymbol{v}_{z,r}^t) \leftarrow$ Back propagation w.r.t $\boldsymbol{v}$ |
|   $w_i \leftarrow w_0^i, v_{z,R}^t \leftarrow 0$ |     Update $\boldsymbol{v}_{z,r+1}^t$ with $\nabla \Omega_z^i(\boldsymbol{v}_{z,r}^t)$ using GD algorithms |
|   **repeat** |     *Repeat the above 7 steps for* |
|     $v_z^t, q^i, q_w^i \leftarrow$ **ClientUpdate** $(i, w_i, \boldsymbol{v}^t)$ |     *the weighted stochastic estimate* |
|     $f(w) = \mathbb{D}_{KL}(q_w^i \| q^i) + \|y - \Phi w^i\|_2^2$ |   $\hat{q^i}(z) \leftarrow \arg \Omega_z^i(\boldsymbol{v}_{z,R}^t), \hat{q_w^i}(z) \leftarrow \arg \Omega_z^i(\boldsymbol{v}_{w,z,R}^t)$ |
|     **Accelerated-IHT**$(f(w))$(Algo 10.7) |   return $\boldsymbol{v}_{z,R}^t, \hat{q^i}(z), \hat{q_w^i}(z)$ |
|   until Stop criteria met | return $\boldsymbol{v}_{z,R}^t$ to the cloud server |

where $b$ and $K$ are minibatch size and Monte Carlo sample size, respectively.

Here $R$ indicates the number iterations after which the clients upload the localized global models to the server. Like (T Dinh et al., 2020), we use an additional parameter $\beta$ in order to make the algorithm converge faster.

## 6 THEORETICAL CONTRIBUTIONS

Here we provide theoretical analysis related to the averaged generalization error for CORESET-PFEDBAYES w.r.t our baseline PFEDBAYES. The main results and derivations of the proofs can be found in the Appendix 9. We first provide certain definitions here.

**Definition 1.** *The Hellinger distance for a particular client $i$ between the estimate likelihood $\mathcal{P}_{\boldsymbol{\theta}}^i$ and the true likelihood $\mathcal{P}^i$ is defined as follows* $d^2(\mathcal{P}_{\boldsymbol{\theta}}^i, \mathcal{P}^i) = \mathbb{E}_{X^i}(1 - e^{-\frac{\left[f_{\boldsymbol{\theta}}^i(X^i) - f^i(X^i)\right]^2}{8\sigma_\epsilon^2}})$

Let $\hat{q^i}(\boldsymbol{\theta}; \boldsymbol{w})$ be the corresponding variational solution for the i-th client's subproblem under the coreset weighted regime and let us define the following term **Generalization Error Term**: $\int_\Theta d^2(\mathcal{P}_{\boldsymbol{\theta}, w}^i, \mathcal{P}^i)\hat{q^i}(\boldsymbol{\theta}; \boldsymbol{w})d\boldsymbol{\theta}$ as the $i$-th client's generalization error.

**Theorem 1.** *The difference in the upper bound incurred in the overall generalization error of* CORESET-PFEDBAYES *as compared w.r.t that of* PFEDBAYES *is always upper bounded by a closed form positive function that depends on the coreset weights and coreset size-* $\mathfrak{S}(\boldsymbol{w}, n_k)$. *generalization error in the original full data setup*

$$\left[\frac{1}{N}\sum_{i=1}^N \int_\Theta d^2(\mathcal{P}_{\boldsymbol{\theta}}^i, \mathcal{P}^i)\hat{q^i}(\boldsymbol{\theta})d\boldsymbol{\theta}\right]_{u.b.} - \left[\frac{1}{N}\sum_{i=1}^N \int_\Theta d^2(\mathcal{P}_{\boldsymbol{\theta},w}^i, \mathcal{P}^i)\hat{q^i}(\boldsymbol{\theta}; \boldsymbol{w})d\boldsymbol{\theta}\right]_{u.b.} \leq \mathfrak{S}(\boldsymbol{w}, n_k)$$

This indicates that the extra estimation error and approximation model incurred by the coreset weighted objective is a direct function of the coreset weight and thus the coreset size and hence can be measured in closed form. Proof. in Appendix 1

**Theorem 2.** *The convergence rate of the generalization error under $L^2$ norm of* CORESET-PFEDBAYES *is minimax optimal up to a logarithmic term (in order $n_k$) for bounded functions ($\beta$-Hölder-smooth functions) $\{f^i\}_{i=1}^N$, $\{f_{\boldsymbol{\theta}}^i\}_{i=1}^N$ and $\{f_{\boldsymbol{\theta}, \boldsymbol{w}}^i\}_{i=1}^N$ where $C_2$, $C_3$ and $\delta'$ are constants (defined in Appendix ) and $\boldsymbol{\Lambda}$ being the intrinsic dimension of each client's data:*

$$\frac{C_F}{N}\sum_{i=1}^N \int_{\boldsymbol{\theta}} \left\|f_{\boldsymbol{\theta}, \boldsymbol{w}}^i(X^i) - f^i(X^i)\right\|_{L^2}^2 \hat{q^i}(\boldsymbol{\theta}; \boldsymbol{w})d\boldsymbol{\theta} \leq C_2 n_k^{-\frac{2\beta}{2\beta+\boldsymbol{\Lambda}}} \log^{2\delta'}(n_k).$$

*and*

$$\inf_{\left\{\|f^i_{\boldsymbol{\theta},\boldsymbol{w}}\|_\infty \leq F\right\}^N_{i=1}\left\{\|f^i\|_\infty \leq F\right\}^N_{i=1}} \frac{C_F}{N} \sum_{i=1}^N \int_{\boldsymbol{\theta}} \left\|f^i_{\boldsymbol{\theta},\boldsymbol{w}}\left(X^i\right) - f^i\left(X^i\right)\right\|^2_{L^2} \hat{q}^i(\boldsymbol{\theta};\boldsymbol{w})d\boldsymbol{\theta} \geq C_3 n_k^{-\frac{2\beta}{2\beta+\Lambda}}$$

*where $n_k$ denotes the coreset size per client dataset and $n$ denotes the original per client dataset*

*size and* $\frac{d^2\left(P^i_{\boldsymbol{\theta},\boldsymbol{w}},P^i\right)}{\left\|f^i_{\boldsymbol{\theta},\boldsymbol{w}}(X^i) - f^i(X^i)\right\|^2_{L^2}} \geq \frac{1-\exp\left(-\frac{4F^2}{8\sigma_\epsilon^2}\right)}{4F^2} \triangleq C_F.$

This indicates that the minimax optimality of the generalization error for CORESET-PFEDBAYES is in logarithmic bounds w.r.t the coreset size $n_k$. Proof. in Appendix 2

**Theorem 3.** *The lower bound (l.b.) incurred for the deviation for the weighted coreset* CORESET-PFEDBAYES *(5) generalization error is always higher than the lower bound of that for*

*the original* PFEDBAYES *objective (1) with a delta difference (**Error I** - **Error II**) as* $\mathcal{O}(n_k^{-\frac{2\beta}{2\beta+\Lambda}})$

$$\underbrace{\left[\sum_{i=1}^N \int_{\Theta} \left\|f^i_{\boldsymbol{\theta},\boldsymbol{w}}\left(X^i\right) - f^i\left(X^i\right)\right\|^2_{L^2} \hat{q}^i(\boldsymbol{\theta},\boldsymbol{w})d\boldsymbol{\theta}\right]_{l.b.}}_{\textit{Coreset weighted objective Generalization Error (\textbf{Error I})}} > \underbrace{\left[\sum_{i=1}^N \int_{\Theta} \left\|f^i_{\boldsymbol{\theta}}\left(X^i\right) - f^i\left(X^i\right)\right\|^2_{L^2} \hat{q}^i(\boldsymbol{\theta})d\boldsymbol{\theta}\right]_{l.b.}}_{\textit{Vanilla objective Generalization Error (\textbf{Error II})}}$$

This simply implies that the generalization error suffers in the case due to limited coreset samples but that is bounded in closed form w.r.t. the coreset sample size. Proof. in Appendix: 3

**Theorem 4.** *The lower bound incurred in the overall generalization error across all $N$ clients of* CORESET-PFEDBAYES *is always higher compared to that of the generalization error in the original full data setup*

$$\left[\frac{1}{N}\sum_{i=1}^N \int_\Theta d^2(\mathcal{P}^i_{\boldsymbol{\theta},w},\mathcal{P}^i)\hat{q}^i(\boldsymbol{\theta};\boldsymbol{w})d\boldsymbol{\theta}\right]_{l.b.} \geq \left[\frac{1}{N}\sum_{i=1}^N \int_\Theta d^2(\mathcal{P}^i_{\boldsymbol{\theta}},\mathcal{P}^i)\hat{q}^i(\boldsymbol{\theta})d\boldsymbol{\theta}\right]_{l.b.}$$

Both Theorem 1 and 4 implies that the overall spread of the Generalization Error Term in case of coreset weighted objective CORESET-PFEDBAYES is much more wider than that of the original PFEDBAYES case. Proof. in Appendix 4

## 7 EXPERIMENTS

Here we perform our experiments to showcase the utility of our method CORESET-PFEDBAYES compared to other baselines like PFEDBAYES and do an in-depth analysis of each of the components involved as follows.

### 7.1 UTILITY OF A-IHT IN VANILLA BAYESIAN CORESETS OPTIMIZATION

In Algorithm 1, since we are employing A-IHT algorithm during coreset updates, we first want to study the utility of applying A-IHT (Accelerated Iterative Thresholding) in a simplistic Bayesian coresets setting using the algorithm proposed by (Huang et al., 2022). For analysis we test the same on Housing Prices 2018 [2]data. Figure 3 showcases the experiments done on the dataset using a riemann linear regression for different coreset sizes of the data (k = 220, 260, 300). As it can be seen the radius capturing the weights w.r.t to the coreset points matches closely at k=300 with that of the true posterior distribution (extreme right), thereby indicating the correctness of approximation and recovery of the true posterior by the A-IHT algorithm.

### 7.2 EXPERIMENTS ON CORESET-PFEDBAYES AGAINST S.O.T.A FEDERATED LEARNING METHODS

Here, we compare the performance of the proposed method CORESET-PFEDBAYES with a variety of baselines such as FedAvg (McMahan et al., 2017), BNFed (Yurochkin et al., 2019), pFedMe

---

[2]https://www.gov.uk/government/statistical-data-sets/price-paid-data-downloads

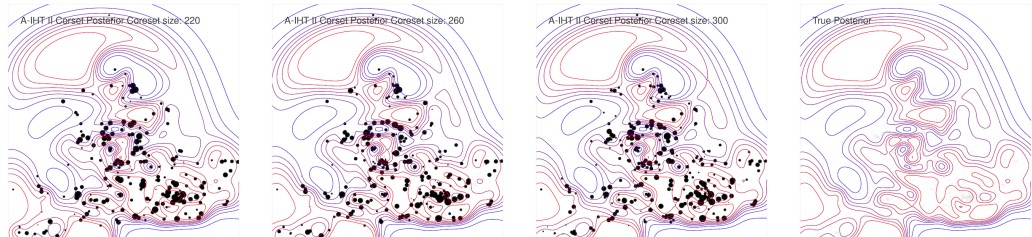

Figure 2: Experiments on Bayesian reimann linear function regression for different settings of core-set size =220,260,300 constructed by Accelerated IHT II. Coreset points are presented as black dots, with their radius indicating assigned weights. Extreme right showcases the true posterior distribution

(T. Dinh et al., 2020), perFedAvg (Fallah et al., 2020b), PFedBayes (Zhang et al., 2022b) based on non-i.i.d. datasets. We generate the non-i.i.d. datasets based on three public benchmark datasets, MNIST (Lecun et al., 1998), FMNIST (Fashion- MNIST) (Xiao et al., 2017) and CIFAR-10 (Krizhevsky et al., 2009). For MNIST, FMNIST and CIFAR-10 datasets, we follow the non-i.i.d. setting strategy in (T Dinh et al., 2020). In our use case we considered total 10 clients each of whom holds a unique local data.

Table 1: Comparative results of personal and global accuracies (in %) across all 7 methods

| Method (Percentage = sampling fraction) | MNIST | | FashionMNIST | | CIFAR | |
|---|---|---|---|---|---|---|
| | Personal Model | Global Model | Personal Model | Global Model | Personal Model | Global Model |
| FedAvg (Full/ 50%) | - | **92.39**(90.60) | - | 85.42(83.90) | - | **79.05**(56.73) |
| BNFed (Full / 50%) | - | 82.95(80.02) | - | 70.1(69.68) | - | 44.37(39.52) |
| pFedMe (Full / 50%) | - | 91.25(89.67) | *92.02*(84.71) | 84.41(83.45) | *77.13*(66.75) | *70.86*(51.18) |
| perFedAvg (Full / 50%) | *98.27* | - | 88.51(84.90) | - | 69.61(52.98) | - |
| PFedBayes (Full / 50%) | **98.79**(90.88) | **97.21**(92.33) | **93.01**(85.95) | **93.30**(82.33) | **83.46**(*73.94*) | 64.40(60.84) |
| RandomSubset (50%) | 80.2 | 88.4 | 87.12 | **90.75** | 48.31 | 61.35 |
| CoreSet-PFedBayes (k = 50%) | *92.48* | *96.3* | *89.55* | *92.7* | 69.66 | **71.5** |

(a) We report accuracies on both global and personal model for the current set of proposed methods across major datasets like **MNIST, CIFAR, FashionMNIST**. **Red** indicates the highest accuracy column-wise. Similarly **Orange** and **Magenta** indicates the 2nd and 3rd best modelwise accuracy. (-) indicates no accuracy reported due to very slow convergence of the corresponding algorithm. **Full indicates training on full dataset and 50% is on using half the data size after randomly sampling 50% of the training set.**

In Table 3, we showcase the accuracy statistics of the corresponding baselines discussed above with our method CORESET-PFEDBAYES across two different configurations **Full Dataset Training** and **50% Training** indicating half of the training samples were selected at random and then the corresponding algorithm was trained). As observed in almost all cases our method CORESET-PFEDBAYES beats PFEDBAYES(random 50% data sampled) by the following margins : +4.87% on MNIST, +8.61% on FashionMNIST, +9.71% and almost on other baselines (random 50% data sampled) and some baselines even when they were trained on full dataset (e.g. our method does better than PFedMe on FashionMNIST). In Appendix 10.2 we showcase the communication complexity of our proposed method.

### 7.3 EXPERIMENTS ON CORESET-PFEDBAYES AGAINST SUBMODULAR SUBSET SELECTION

In order to showcase the advantage of model-centric subset selection methods over traditional data-centric(model-agnostic) methods like our proposed work CORESET-PFEDBAYES which takes into account matching the client distribution under coreset setting $q^i(\theta; w)$ to that in normal setting $q^i(\theta)$. Hence, we compare our proposed method CORESET-PFEDBAYES against **submodular based functions** (*See Appendix 7 for definition*) (specifically, *diversity* based submodular functions as the aim is to select a subset of data points that are most diverse). The discussion on some of the common diversity functions and their properties with regard to monotonicity and submodularity are provided in Appendix 10.5. Each of the medical datasets consists of 3 classes out of which 1 class (Normal is kept as common) between 2 clients and the data about the other two classes is distributed separately to the 2 clients. Our aim here is to deploy our proposed method in this setting and compare against submodular based subset selection approaches.

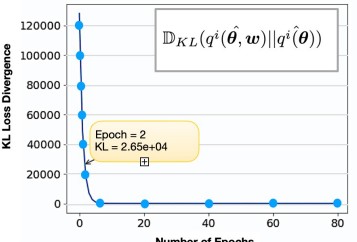

KL Divergence between the corset weighted client side distribution and normal client side distribution

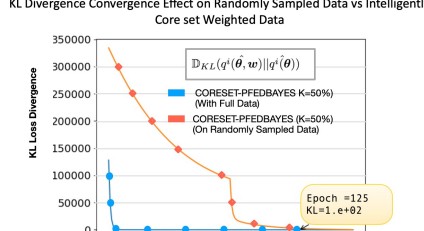

KL Divergence Convergence Effect on Randomly Sampled Data vs Intelligently Core set Weighted Data

(a) The optimal distribution parameters learnt after taking a coreset of the data tend to match up with parameters, when trained on the full dataset after few epochs resulting in decreasing KL divergence score. This is in line with our initial hypothesis 4.3

(b) We are comparing a random subset selection of the data vs when a subset is selected using our method, the early convergence of the KL-divergence shows the better performance of our method. When trained on a random subset of data, the model takes more epochs to converge.

Figure 3: KL Divergence Plot over Number of Epochs (MNIST Dataset)

More details regarding the dataset description along with the experimental configurations can be found under Appendix. Note for each of the submodular function baselines, the data was sampled using a submodular function optimization strategy post which FedAvg was applied as the Federated Learning algorithm.

Table 2: Comparative results of classwise global accuracies of all 9 methods on **3 different medical datasets** and **2 clients**

| Method (Percentage = sampling fraction) | COVID-19 Radiography Database | | | APTOS 2019 Blindness Detection | | | OCTMNIST Dataset | | |
|---|---|---|---|---|---|---|---|---|---|
| | Normal X-ray | COVID X-ray | Lung Opacity X-ray | Normal Retina | Mild Diabetic Retinopathy | Severe Diabetic Retinopathy | Normal Retina | DME | Drusen |
| **Vanilla FedAvg (Full)** | **0.914 ± 0.007** | **0.924 ± 0.005** | **0.898 ± 0.007** | **0.968 ± 0.023** | **0.927 ± 0.019** | **0.853 ± 0.004** | **0.908 ± 0.026** | 0.837 ± 0.103 | **0.855 ± 0.092** |
| **PFEDBAYES(Full)** | **0.953 ± 0.006** | **0.938 ± 0.004** | **0.902 ± 0.011** | **0.951 ± 0.057** | **0.941 ± 0.052** | **0.911 ± 0.028** | **0.926 ± 0.013** | 0.851 ± 0.021 | **0.874 ± 0.012** |
| **Independent Learning (Full)** | **0.898 ± 0.001** | **0.869 ± 0.002** | **0.884 ± 0.003** | **0.945 ± 0.025** | 0.877 ± 0.049 | 0.830 ± 0.053 | **0.890 ± 0.073** | 0.798 ± 0.076 | **0.890 ± 0.041** |
| **RandomSub FedAvg (50%)** | 0.892 ± 0.024 | 0.670 ± 0.059 | 0.583 ± 0.033 | 0.918 ± 0.047 | 0.835 ± 0.091 | 0.832 ± 0.021 | 0.811 ± 0.070 | 0.753 ± 0.089 | 0.805 ± 0.068 |
| **LogDet FedAvg (50%)** | 0.887 ± 0.046 | 0.838 ± 0.086 | 0.810 ± 0.041 | 0.918 ± 0.027 | **0.885 ± 0.082** | **0.850 ± 0.057** | **0.842 ± 0.046** | **0.897 ± 0.039** | 0.845 ± 0.068 |
| **DispSum FedAvg (50%)** | **0.907 ± 0.015** | **0.925 ± 0.049** | **0.812 ± 0.086** | **0.945 ± 0.043** | **0.890 ± 0.095** | **0.852 ± 0.061** | 0.834 ± 0.044 | **0.887 ± 0.082** | **0.863 ± 0.094** |
| **DispMin FedAvg (50%)** | 0.866 ± 0.018 | 0.780 ± 0.045 | 0.751 ± 0.069 | **0.963 ± 0.021** | 0.851 ± 0.067 | 0.765 ± 0.033 | 0.831 ± 0.011 | **0.892 ± 0.066** | 0.835 ± 0.085 |
| **CORESET-PFEDBAYES (50%)** | **0.932 ± 0.003** | **0.919 ± 0.013** | **0.871 ± 0.025** | 0.921 ± 0.016 | **0.894 ± 0.029** | **0.886 ± 0.017** | **0.916 ± 0.042** | 0.805 ± 0.008 | **0.816 ± 0.011** |

(a) We report classwise accuracies for the current set of proposed methods for all 3 medical datasets. **Red** indicates the highest value in accuracy column-wise (i.e. for a particular class for a dataset across all 9 baselines). Similarly **Orange** and **Magenta** indicates the 2nd and 3rd best classwise accuracy. **Colors for Vanilla FedAvg, PFEDBAYES , CORESET-PFEDBAYES are grouped together** to primarily compare against subset selection strategies

As observed in Table 2 CORESET-PFEDBAYES performs better than submodular based approaches on average across Covid-19 and APTOS. This is promising as it indicates model centric subset selection is much more useful in terms of performance than model agnostic subset selection methods.

## 8    CONCLUSION & FUTURE WORK

In this work we proposed several novel objective formulations that draw and synthesize from previous works of two separate domains: coreset optimization and federated learning. Through our extensive experimentations, our proposed method showcases significant gains over traditional federated learning approaches and submodularity based optimization functions followed by Federated Learning. We also showcased through theoretical analysis, how the average generalization error is minimax optimal upto logarithm bounds and how that estimation and approximation error compares against PFEDBAYES. In future we want to look into how client-wise data distribution affects the current scheme and how we can make the model more robust towards adversarial attacks from (say) skewed data distribution over client-side model parameters. Also, the interplay of how such coreset weights can affect model updates in a privacy-preserving manner is somewhat interesting to explore further.

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

# Supplementary Material: Bayesian Coreset Optimization for Personalized Federated Learning

In this supplementary material we discuss extensively on the proofs. involved for the theoretical analysis for CORESET-PFEDBAYES along with more fine-grained experimental details and corresponding baselines.

## 9 PROOFS

Here we discuss the proofs involved with particular propositions and theorems specified in the Theoretical Contributions of this paper. Utilising the assumptions taken in Zhang et al. (2022b), Polson & Ročková (2018) we consider the analysis for equal-width Bayesian Neural network.

**Assumption 1**: The widths of the neural network are equal width i.e. $s_i = M$.

**Assumption 2**: Each individual client $i \in [N]$ has equal coreset size of samples $n_k < n$.

**Assumption 3**: Parameters $s_0, n$ (*total client dataset size*) $n_k$ ( *coreset client dataset size*), $M$, $L$ (number of DNN layers as per Section 3) are large enough such that the sequence $\sigma_n^2$ is bounded as follows

$$\sigma_n^2 = \frac{T}{8n}A \leq \Omega^2,$$

where $\tau = \Omega M$ and

$$A = \log^{-1}\left(3s_0 M\right) \cdot (2\tau)^{-2(L+1)}$$
$$\left[\left(s_0 + 1 + \frac{1}{\tau - 1}\right)^2 + \frac{1}{(2\tau)^2 - 1} + \frac{2}{(2\tau - 1)^2}\right]^{-1}.$$

Here $T$ indicates the total number of parameters as defined in Section 3

Similarly, utilising the coreset regime, we have the following:

$$\sigma_{n_k}^2 = \frac{T}{8n_k}A \leq \Omega^2,$$

Since $n_k << n$, hence $\sigma_{n_k}^2 >> \sigma_n^2$

**Assumption 4**: We consider 1-Lipschitz continuous activation function $\sigma(\bullet)$

We also define here a few terms as defined in (Zhang et al., 2022b) which would be useful for our following proof proposals as well.

**Definition 2.** *Preliminaries and Definitions required for theoretical proofs under* PFEDBAYES

$$d^2(\mathcal{P}_{\boldsymbol{\theta}}^i, \mathcal{P}^i) = \mathbb{E}_{X^i}\left(1 - e^{-\frac{\left[f_{\boldsymbol{\theta}}^i(X^i) - f^i(X^i)\right]^2}{8\sigma_\epsilon^2}}\right)$$
$$r_n = ((L+1)T/n)\log M + (T/n)\log\left(s_0\sqrt{n/T}\right)$$
$$\xi_n^i = \inf_{\boldsymbol{\theta} \in \Theta(L, \boldsymbol{S}), \|\boldsymbol{\theta}\|_\infty \leq \Omega} \left\|f_{\boldsymbol{\theta}}^i - f^i\right\|_\infty^2,$$
$$\varepsilon_n = n^{-\frac{1}{2}}\sqrt{(L+1)T\log M + T\log\left(s_0\sqrt{n/T}\right)}\log^\delta(n) = \sqrt{r_n}\log^\delta(n),$$

*where $\delta > 1$*

Here $r_n$ indicates the variational error incurred due to the Bayesian approximation to the true posterior distribution in Equation 1 and $\xi_n^i$ indicates the approximation error incurred during regression w.r.t the actual function to be learnt.

Similarly for the coreset size $n_k$ we define the following:

**Definition 3.** *Preliminaries and Definitions required for theoretical proofs under* CORESET-PFEDBAYES

$$d^2(\mathcal{P}_{\boldsymbol{\theta},\boldsymbol{w}}^i, \mathcal{P}^i) = \mathbb{E}_{X^i}(1 - e^{-\frac{\left[f_{\boldsymbol{\theta},\boldsymbol{w}}^i(X^i) - f^i(X^i)\right]^2}{8\sigma_\epsilon^2}})$$

$$\xi_{n_k}^i = \inf_{\boldsymbol{\theta} \in \boldsymbol{\Theta}(L,\boldsymbol{S}), \|\boldsymbol{\theta}\|_\infty \leq \Omega} \left\| f_{\boldsymbol{\theta},\boldsymbol{w}}^i - f^i \right\|_\infty^2$$

$$r_{n_k} = ((L+1)T/n_k)\log M + (T/n_k)\log\left(s_0\sqrt{n_k/T}\right)$$

$$\varepsilon_{n_k} = n_k^{-\frac{1}{2}}\sqrt{(L+1)T\log M + T\log\left(s_0\sqrt{n_k/T}\right)}\log^\delta(n_k) = \sqrt{r_{n_k}}\log^\delta(n_k)$$

**Lemma 1.** *The Hellinger Distance from Definition 1 is symmetrical in its arguments* $\mathcal{P}_{\boldsymbol{\theta}}^i$ *and* $\mathcal{P}^i$.

*Proof.* It is easy to show that,

$$d^2(\mathcal{P}_{\boldsymbol{\theta}}^i, \mathcal{P}^i) = \mathbb{E}_{X^i}(1 - e^{-\frac{[f_{\boldsymbol{\theta}}^i(X^i) - f^i(X^i)]^2}{8\sigma_\epsilon^2}}) \tag{10}$$

$$= \mathbb{E}_{X^i}(1 - e^{-\frac{[f^i(X^i) - f_{\boldsymbol{\theta}}^i(X^i)]^2}{8\sigma_\epsilon^2}}) \tag{11}$$

$$= d^2(\mathcal{P}^i, \mathcal{P}_{\boldsymbol{\theta}}^i) \tag{12}$$

$\square$

**PROOF. OF THEOREM 1**

**Theorem 1.** *The difference in the upper bound incurred in the overall generalization error of* CORESET-PFEDBAYES *as compared w.r.t that of* PFEDBAYES *is always upper bounded by a closed form positive function that depends on the coreset weights and coreset size-* $\mathfrak{S}(\boldsymbol{w}, n_k)$. *generalization error in the original full data setup*

$$\left[\frac{1}{N}\sum_{i=1}^N \int_\Theta d^2(\mathcal{P}_{\boldsymbol{\theta}}^i, \mathcal{P}^i)\hat{q}^i(\boldsymbol{\theta})d\boldsymbol{\theta}\right]_{u.b.} - \left[\frac{1}{N}\sum_{i=1}^N \int_\Theta d^2(\mathcal{P}_{\boldsymbol{\theta},w}^i, \mathcal{P}^i)\hat{q}^i(\boldsymbol{\theta};\boldsymbol{w})d\boldsymbol{\theta}\right]_{u.b.} \leq \mathfrak{S}(\boldsymbol{w}, n_k)$$

*Proof.* Let us define $\log\eta(P_{\boldsymbol{\theta}}^i, P^i) = l_n(P_{\boldsymbol{\theta}}^i, P^i)/\zeta + nd^2(P_{\boldsymbol{\theta}}^i, P^i)$.

Using Theorem 3.1 of Pati et al. (2018) with probability at most $e^{-Cn_k\varepsilon_{n_k}^2}$, where $C$ is a constant, with high probability for CORESET-PFEDBAYES we have

$$\int_\Theta \eta(\mathcal{P}_{\boldsymbol{\theta},\boldsymbol{w}}^i, \mathcal{P}^i)z^*(\boldsymbol{\theta})d\boldsymbol{\theta} \leq e^{Cn_k\varepsilon_{n_k}^2} \tag{13}$$

Similarly with high probability at most $e^{-Cn\varepsilon_n^2}$ for the vanilla PFEDBAYES

$$\int_\Theta \eta(\mathcal{P}_{\boldsymbol{\theta}}^i, \mathcal{P}^i)z^*(\boldsymbol{\theta})d\boldsymbol{\theta} \leq e^{Cn\varepsilon_n^2} \tag{14}$$

Using Lemma A.1 from Zhang et al. (2022b) we know that for any probability measure $\mu$ and any measurable function $h$ with $e^h \in L_1(\mu)$,

$$\log\int e^{h(\eta)}\mu(d\eta) = \sup_\rho\left[\int h(\eta)\rho(d\eta) - \mathbb{D}_{KL}(\rho\|\mu)\right]$$

Further, we let $l_n\left(P^i, P^i_{\boldsymbol{\theta}}\right)$ is the log-likelihood ratio of $P^i$ and $P^i_{\boldsymbol{\theta}}$

$$l_n\left(P^i, P^i_{\theta}\right) = \log \frac{\mathcal{P}^i\left(\boldsymbol{D}^i\right)}{\mathcal{P}^i_{\theta}\left(\boldsymbol{D}^i\right)}.$$

Hence,

$$\begin{aligned} nd^2(P^i_{\boldsymbol{\theta}}, P^i) &= l_n(P^i_{\boldsymbol{\theta}}, P^i)/\zeta - \log \eta(P^i_{\boldsymbol{\theta}}, P^i) \\ &= l_n(P^i, P^i_{\boldsymbol{\theta}})/\zeta - \log \eta(P^i, P^i_{\theta}) \quad \text{since } d^2(P^i_{\boldsymbol{\theta}}, P^i) = d^2(P^i, P^i_{\boldsymbol{\theta}}) \text{ from Lemma 1} \end{aligned}$$

This follows from 9

Similarly, for the weighted likelihood based Hellinger Distance,

$$n_k d^2(P^i_{\boldsymbol{\theta},\boldsymbol{w}}, P^i) = l_n(P^i_{\boldsymbol{\theta},\boldsymbol{w}}, P^i)/\zeta - \log \eta(P^i_{\boldsymbol{\theta},\boldsymbol{w}}, P^i) \tag{15}$$

By using Lemma A.1 with $h(\eta) = \log \eta\left(P^i_{\boldsymbol{\theta}}, P^i\right)$, $\mu = z^\star(\boldsymbol{\theta})$ and $\rho = \hat{q}^i(\boldsymbol{\theta})$, we obtain

$$\begin{aligned} \int_{\Theta} d^2\left(P^i_{\boldsymbol{\theta}}, P^i\right) \hat{q}^i(\boldsymbol{\theta}) d\boldsymbol{\theta} &\leq \frac{1}{n}\left[\frac{1}{\zeta}\int_{\Theta} l_n\left(P^i, P^i_{\boldsymbol{\theta}}\right)\hat{q}^i(\boldsymbol{\theta})d\boldsymbol{\theta} + \mathbb{D}_{KL}\left(\hat{q}^i(\boldsymbol{\theta})\|z^\star(\boldsymbol{\theta})\right) + \log\int_{\Theta}\eta\left(P^i_{\boldsymbol{\theta}}, P^i\right)z^\star(\boldsymbol{\theta})d\boldsymbol{\theta}\right] \\ &\leq \frac{1}{n}\left[\frac{1}{\zeta}\int_{\Theta} l_n\left(P^i, P^i_{\boldsymbol{\theta}}\right)\hat{q}^i(\boldsymbol{\theta})d\boldsymbol{\theta} + \mathbb{D}_{KL}\left(\hat{q}^i(\boldsymbol{\theta})\|z^\star(\boldsymbol{\theta})\right)\right] + C\varepsilon^2_n \end{aligned}$$

$$\int_{\Theta} d^2(\mathcal{P}^i_{\boldsymbol{\theta},\boldsymbol{w}}, \mathcal{P}^i)q^i(\hat{\boldsymbol{\theta}}, \boldsymbol{w})d\boldsymbol{\theta} \leq \frac{1}{n_k}\left[\frac{1}{\zeta}\int_{\Theta} l_n\left(P^i, P^i_{\boldsymbol{\theta},\boldsymbol{w}}\right)q^i(\hat{\boldsymbol{\theta}}; \boldsymbol{w})d\boldsymbol{\theta} + \mathbb{D}_{KL}\left(q^i(\hat{\boldsymbol{\theta}}; \boldsymbol{w})\|z^\star(\boldsymbol{\theta})\right)\right] + C\varepsilon^2_{n_k}$$

Utilising analysis under Supplementary in (Bai et al., 2020), there exists an upper bound for the term

$$\int_{\Theta} l_n\left(P^i, P^i_{\boldsymbol{\theta}}\right)\hat{q}^i(\boldsymbol{\theta})d\boldsymbol{\theta} \leq C''(nr_n + n\xi^i_n) \tag{16}$$

Lemma 2 from (Zhang et al., 2022b) provides the upper bound for the KL divergence term

$$\mathbb{D}_{KL}\left(\hat{q}^i(\boldsymbol{\theta})\|z^\star(\boldsymbol{\theta})\right) \leq C'(nr_n) \tag{17}$$

Therefore we can write the following expression that captures the weighted Hellinger distance displacement given in our coreset framework CORESET-PFEDBAYES as compared to PFEDBAYES

$$\frac{1}{N} \sum_{i=1}^{N} \int_{\Theta} d^2(\mathcal{P}_{\boldsymbol{\theta}}^i, \mathcal{P}^i) q^i(\hat{\boldsymbol{\theta}}) d\boldsymbol{\theta} - \frac{1}{N} \sum_{i=1}^{N} \int_{\Theta} d^2(\mathcal{P}_{\boldsymbol{\theta},\boldsymbol{w}}^i, \mathcal{P}^i) q^i(\hat{\boldsymbol{\theta}}, \boldsymbol{w}) d\boldsymbol{\theta}$$

$$\leq \frac{1}{N} \sum_{i=1}^{N} \frac{1}{n} \left[ \frac{1}{\zeta} \int_{\Theta} l_n \left( P^i, P_{\boldsymbol{\theta}}^i \right) \hat{q}^i(\boldsymbol{\theta}) d\boldsymbol{\theta} + \mathbb{D}_{KL} \left( \hat{q}^i(\boldsymbol{\theta}) \| z^\star(\boldsymbol{\theta}) \right) \right] + C\varepsilon_n^2 -$$

$$\frac{1}{N} \sum_{i=1}^{N} \frac{1}{n_k} \left[ \frac{1}{\zeta} \int_{\Theta} l_n \left( P^i, P_{\boldsymbol{\theta},w}^i \right) q^i(\hat{\boldsymbol{\theta}}; \boldsymbol{w}) d\boldsymbol{\theta} + \mathbb{D}_{KL} \left( q^i(\hat{\boldsymbol{\theta}}; \boldsymbol{w}) \| z^\star(\boldsymbol{\theta}) \right) \right] - C\varepsilon_{n_k}^2$$

Using Eq:(17) and Eq:(16)

$$\leq C\varepsilon_n^2 - C\varepsilon_{n_k}^2 + n\left( C' \zeta r_n + \frac{C''}{N} \sum_{i=1}^{N} \xi_n^i \right) - n_k\left( C' \zeta r_{n_k} + \frac{C''}{N} \sum_{i=1}^{N} \xi_{n_k}^i \right)$$

$$\leq C(\varepsilon_n^2 - \varepsilon_{n_k}^2) + \zeta C'(nr_n - n_k r_{n_k}) + \frac{C''}{N} \sum_{i=1}^{N}(n\xi_n^i - n_k \xi_{n_k}^i)$$

$$= C \underbrace{(\varepsilon_n^2 - \varepsilon_{n_k}^2)}_{\substack{\text{Estimation error} \\ \text{Type I \textbf{Drift}}}} + \zeta C' \underbrace{(nr_n - n_k r_{n_k})}_{\substack{\text{Estimation error} \\ \text{Type II \textbf{Drift}}}} + \frac{C''}{N} \underbrace{\sum_{i=1}^{N}(n\xi_n^i - n_k \xi_{n_k}^i)}_{\text{Approximation Error \textbf{Drift}}}$$

$$= \underbrace{\mathfrak{S}(\boldsymbol{w}, n_k)}_{\geq 0}$$

Where $\mathfrak{S}(\boldsymbol{w}, n_k) = C(\varepsilon_n^2 - \varepsilon_{n_k}^2) + \zeta C'(nr_n - n_k r_{n_k}) + \frac{C''}{N} \sum_{i=1}^{N}(n\xi_n^i - n_k \xi_{n_k}^i)$ where each of the coefficients of the closed form function are constants related to $s_0, \beta, \boldsymbol{\Lambda}, L, M, \zeta$ and $n_k$

Using Lemma 2, 3 and with suitable assumptions on the Approximation drift error such that we see that each of the individual error terms are positive, there by indicating $\mathfrak{S}(\boldsymbol{w}, n_k) \geq 0$ $\qquad \square$

**Lemma 2.** *The Estimation error Type II **Drift** is a positive quantity i.e. $nr_n > n_k r_{n_k}$.*

*Proof.* By Definition,

$$r_n = ((L+1)T/n) \log M + (T/n) \log \left( s_0 \sqrt{n/T} \right)$$

and

$$r_{n_k} = ((L+1)T/n_k) \log M + (T/n_k) \log \left( s_0 \sqrt{n_k/T} \right)$$

Hence

$$\frac{r_n}{r_{n_k}} = \frac{n_k}{n} \times \frac{((L+1)T) \log M + (T) \log \left( s_0 \sqrt{n/T} \right)}{((L+1)T) \log M + (T) \log \left( s_0 \sqrt{n_k/T} \right)}$$

$$\frac{r_n}{r_{n_k}} = \frac{n_k}{n} \times \frac{(L+1) \log M + \log \left( s_0/\sqrt{T} \right) + \log \left( \sqrt{n} \right)}{(L+1) \log M + \log \left( s_0/\sqrt{T} \right) + \log \left( \sqrt{n_k} \right)}$$

Considering $(L+1) \log M + \log \left( s_0/\sqrt{T} \right)$ as a constant $\mathfrak{G}$ we have

$$\frac{r_n}{r_{n_k}} = \frac{n_k}{n} \times \frac{\mathfrak{G} + \log(\sqrt{n})}{\mathfrak{G} + \log(\sqrt{n_k})}$$

Thus,

$$\frac{n r_n}{n_k r_{n_k}} = \frac{\mathfrak{G} + \log(\sqrt{n})}{\mathfrak{G} + \log(\sqrt{n_k})}$$

It is clear since $\log(\bullet)$ is an increasing function for $n > n_k$ we have $n r_n > n_k r_{n_k}$. $\qquad\square$

**Lemma 3.** *The Estimation error Type  1 **Drift** is a positive quantity i.e. $\varepsilon_n^2 > \varepsilon_{n_k}^2$.*

*Proof.* From the definition under Assumption 3

$$\varepsilon_{n_k} = n_k^{-\frac{1}{2}} \sqrt{(L+1)T \log M + T \log\left(s_0 \sqrt{n_k/T}\right)} \log^\delta(n_k) = \sqrt{r_{n_k}} \log^\delta(n_k)$$

Hence $\varepsilon_{n_k}^2 = r_{n_k} \log^{2\delta}(n_k)$. Similarly, $\varepsilon_n^2 = r_n \log^{2\delta}(n)$

$$\frac{\varepsilon_n^2}{\varepsilon_{n_k}^2} = \frac{r_n \log^{2\delta}(n)}{r_{n_k} \log^{2\delta}(n_k)} = \frac{n r_n \frac{\log^{2\delta}(n)}{n}}{n_k r_{n_k} \frac{\log^{2\delta}(n_k)}{n_k}}$$

From Lemma 2 we know that $n r_n > n_k r_{n_k}$, hence

$$\frac{\varepsilon_n^2}{\varepsilon_{n_k}^2} > \frac{\frac{\log^{2\delta}(n)}{n}}{\frac{\log^{2\delta}(n_k)}{n_k}} > 1$$

This follows due to the increasing nature of the function.

$\qquad\square$

## PROOF. OF THEOREM 2

**Theorem 2.** *The convergence rate of the generalization error under $L^2$ norm of* CORESET-PFEDBAYES *is minimax optimal up to a logarithmic term (in order $n_k$) for bounded functions ($\beta$-Hölder-smooth functions) $\{f^i\}_{i=1}^N$, $\{f_\theta^i\}_{i=1}^N$ and $\{f_{\theta,w}^i\}_{i=1}^N$ where $C_2$, $C_3$ and $\delta'$ are constants and $\mathbf{\Lambda}$ being the intrinsic dimension of each client's data:*

$$\frac{C_F}{N} \sum_{i=1}^N \int_{\boldsymbol{\theta}} \left\| f_{\boldsymbol{\theta},\boldsymbol{w}}^i\left(X^i\right) - f^i\left(X^i\right) \right\|_{L^2}^2 \hat{q}^i(\boldsymbol{\theta}; \boldsymbol{w}) d\boldsymbol{\theta} \leq C_2 n_k^{-\frac{2\beta}{2\beta+\Lambda}} \log^{2\delta'}(n_k).$$

*and*

$$\inf_{\left\{\|f_{\boldsymbol{\theta},\boldsymbol{w}}^i\|_\infty \leq F\right\}_{i=1}^N \left\{\|f^i\|_\infty \leq F\right\}_{i=1}^N} \frac{C_F}{N} \sum_{i=1}^N \int_{\boldsymbol{\theta}} \left\| f_{\boldsymbol{\theta},\boldsymbol{w}}^i\left(X^i\right) - f^i\left(X^i\right) \right\|_{L^2}^2 \hat{q}^i(\boldsymbol{\theta}; \boldsymbol{w}) d\boldsymbol{\theta} \geq C_3 n_k^{-\frac{2\beta}{2\beta+\Lambda}}$$

*where $n_k$ denotes the coreset size per client dataset and $n$ denotes the original per client dataset size and* $\dfrac{d^2\left(P_{\boldsymbol{\theta},\boldsymbol{w}}^i, P^i\right)}{\left\| f_{\boldsymbol{\theta},\boldsymbol{w}}^i(X^i) - f^i(X^i) \right\|_{L^2}^2} \geq \dfrac{1 - \exp\left(-\frac{4F^2}{8\sigma_\epsilon^2}\right)}{4F^2} \triangleq C_F.$

We present the choice of $T$ for a typical class of functions. We already assumed that $\{f^i\}$ are $\beta$-Hölder-smooth functions (Definition 4. (Nakada & Imaizumi, 2020)) and the intrinsic dimension of data is $\mathbf{\Lambda}$.

From our above theorem result from Theorem: 1 we say the following:

$$\frac{1}{N}\sum_{i=1}^{N}\int_{\Theta}d^2(\mathcal{P}_{\boldsymbol{\theta},\boldsymbol{w}}^i,\mathcal{P}^i)\hat{q}^i(\boldsymbol{\theta};\mathbf{w})d\boldsymbol{\theta} \leq C\varepsilon_{n_k}^2 + C'r_{n_k} + \frac{C''}{N\zeta}\sum_{i=1}^{N}\xi_{n_k}^i \tag{18}$$

Utilising Corollary 6 in (Nakada & Imaizumi, 2020) , the approximation error is upper-bounded as follows

$$\left\|f_{\boldsymbol{\theta},\boldsymbol{w}}^i - f^i\right\|_{\infty} \leq C_0 T^{-\frac{\beta}{\Lambda}}$$

where $C_0 > 0$ is a constant related to $s_0, \beta$ and $\mathbf{\Lambda}$

Thus from the above definitions 2 and 3, we have the following

$$\xi_n^i, \xi_{n_k}^i \leq C_0 T^{-\frac{2\beta}{\Lambda}}, i = 1, \ldots, N$$

Utilising the above upper bound in 18 and substituting $T = C_1 n^{\frac{\Lambda}{2\beta+\Lambda}}$, we get

$$\frac{1}{N}\sum_{i=1}^{N}\int_{\Theta}d^2(\mathcal{P}_{\boldsymbol{\theta},\boldsymbol{w}}^i,\mathcal{P}^i)\hat{q}^i(\boldsymbol{\theta};\mathbf{w})d\boldsymbol{\theta} \leq C\varepsilon_{n_k}^2 + C'r_{n_k} + \frac{C''}{N\zeta}\sum_{i=1}^{N}C_0 T^{-\frac{2\beta}{\Lambda}}$$

$$\leq Cr_{n_k}\log^{2\delta}(n_k) + C'r_{n_k} + \frac{C''}{N\zeta}\sum_{i=1}^{N}C_0 T^{-\frac{2\beta}{\Lambda}} \because \varepsilon_{n_k}^2 = r_{n_k}\log^{2\delta}(n_k)$$

$$\leq C_2 n_k^{-\frac{2\beta}{2\beta+\Lambda}}\log^{2\delta'}(n_k)\big[\text{ substituting } T \text{ in } r_{n_k}\big]$$

where $\delta' > \delta > 1$, and $C_1, C_2 > 0$ are constants related to $s_0, \beta, \mathbf{\Lambda}, L, M, \zeta$ and $n_k$.

Similar to Theorem 1.1 from (Bai et al., 2020) and Theorem 1 from (Zhang et al., 2022b) norm, we can write the following

$$\frac{C_F}{N}\sum_{i=1}^{N}\int_{\Theta}\left\|f_{\boldsymbol{\theta},\boldsymbol{w}}^i\left(X^i\right) - f^i\left(X^i\right)\right\|_{L^2}^2\hat{q}^i(\boldsymbol{\theta},\boldsymbol{w})d\boldsymbol{\theta}$$

$$\leq \frac{1}{N}\sum_{i=1}^{N}\int_{\Theta}d^2\left(P_{\boldsymbol{\theta},\boldsymbol{w}}^i,P^i\right)\hat{q}^i(\boldsymbol{\theta};\boldsymbol{w})d\boldsymbol{\theta}$$

$$\leq C_2 n_k^{-\frac{2\beta}{2\beta+\Lambda}}\log^{2\delta'}(n_k).$$

Now, using the minimax lower bound under $L^2$ norm in Theorem 8 of (Nakada & Imaizumi, 2020), we see that for coreset regime the same formulation holds similar to our original setting as shown in (Zhang et al., 2022b)

$$\inf_{\left\{\|f_{\boldsymbol{\theta},\boldsymbol{w}}^i\|_{\infty} \leq F\right\}_{i=1}^{N}\left\{\|f^i\|_{\infty} \leq F\right\}_{i=1}^{N}}\frac{C_F}{N}\sum_{i=1}^{N}\int_{\boldsymbol{\theta}}\left\|f_{\boldsymbol{\theta},\boldsymbol{w}}^i\left(X^i\right) - f^i\left(X^i\right)\right\|_{L^2}^2\hat{q}^i(\boldsymbol{\theta};\boldsymbol{w})d\boldsymbol{\theta} \geq C_3 n_k^{-\frac{2\beta}{2\beta+\Lambda}}$$

where $C_3 > 0$ is a constant.

Combining the above two equations, the convergence rate of the generalization error of the coreset weighted objective is minimax optimal upto a logarithmic term for bounded functions $\{f_{\boldsymbol{\theta},\boldsymbol{w}}^i\}_{i=1}^{N}$ and $\{f_{\boldsymbol{\theta}}^i\}_{i=1}^{N}$.

**PROOF. OF THEOREM** 3

**Theorem 3.** *The lower bound (l.b.) incurred for the deviation for the weighted coreset* CORESET-PFEDBAYES *(5) generalization error is always higher than the lower bound of that for the original* PFEDBAYES *objective (1) with a delta difference (**Error I** - **Error II**) as $\mathcal{O}(n_k^{-\frac{2\beta}{2\beta+\Lambda}})$*

$$\left[\sum_{i=1}^{N}\int_{\Theta}\left\|f_{\boldsymbol{\theta},\boldsymbol{w}}^{i}\left(X^{i}\right)-f^{i}\left(X^{i}\right)\right\|_{L^{2}}^{2}\hat{q}^{i}(\boldsymbol{\theta},\boldsymbol{w})d\boldsymbol{\theta}\right]_{l.b.} > \left[\sum_{i=1}^{N}\int_{\Theta}\left\|f_{\boldsymbol{\theta}}^{i}\left(X^{i}\right)-f^{i}\left(X^{i}\right)\right\|_{L^{2}}^{2}\hat{q}^{i}(\boldsymbol{\theta})d\boldsymbol{\theta}\right]_{l.b.}$$

$$\underbrace{\phantom{xxxxxxxxxxxxxxxxxxxxxxxxxxxxxxxxxxxxxxxxxxxx}}_{\textit{Coreset weighted objective Generalization Error (\textbf{Error I})}} \qquad \underbrace{\phantom{xxxxxxxxxxxxxxxxxxxxxxxxxxxxxxxx}}_{\textit{Vanilla objective Generalization Error (\textbf{Error II})}}$$

*Proof.* As we know $n_k < n$ hence $C_3 n_k^{-\frac{2\beta}{2\beta+\Lambda}} > C_3 n^{-\frac{2\beta}{2\beta+\Lambda}}$ ($\because C_3$ is a constant independent of $n$ or $n_k$), which therefore means that inequality holds in the lower bound (l.b.) of the two expressions (shown by the previous proposition 2).

$$\left[\sum_{i=1}^{N}\int_{\boldsymbol{\theta}}\left\|f_{\boldsymbol{\theta},\boldsymbol{w}}^{i}\left(X^{i}\right)-f^{i}\left(X^{i}\right)\right\|_{L^{2}}^{2}\hat{q}^{i}(\boldsymbol{\theta},\boldsymbol{w})d\boldsymbol{\theta}\right]_{l.b.} > \left[\sum_{i=1}^{N}\int_{\boldsymbol{\theta}}\left\|f_{\boldsymbol{\theta}}^{i}\left(X^{i}\right)-f^{i}\left(X^{i}\right)\right\|_{L^{2}}^{2}\hat{q}^{i}(\boldsymbol{\theta})d\boldsymbol{\theta}\right]_{l.b.}$$

Let us denote $\boldsymbol{\Delta}_{deviation}^{l.b}$ as follows

$$\boldsymbol{\Delta}_{deviation}^{l.b} = \left[\sum_{i=1}^{N}\int_{\boldsymbol{\theta}}\left\|f_{\boldsymbol{\theta},\boldsymbol{w}}^{i}\left(X^{i}\right)-f^{i}\left(X^{i}\right)\right\|_{L^{2}}^{2}\hat{q}^{i}(\boldsymbol{\theta},\boldsymbol{w})d\boldsymbol{\theta}\right]_{l.b.} - \left[\sum_{i=1}^{N}\int_{\boldsymbol{\theta}}\left\|f_{\boldsymbol{\theta}}^{i}\left(X^{i}\right)-f^{i}\left(X^{i}\right)\right\|_{L^{2}}^{2}\hat{q}^{i}(\boldsymbol{\theta})d\boldsymbol{\theta}\right]_{l.b.}$$

And the $\boldsymbol{\Delta}_{deviation}^{l.b.}$ term is given by $\left(C_3 n_k^{-\frac{2\beta}{2\beta+\Lambda}} - C_3 n^{-\frac{2\beta}{2\beta+\Lambda}}\right) \approx \mathcal{O}(n_k^{-\frac{2\beta}{2\beta+\Lambda}})$. $\qquad\square$

**PROOF. OF THEOREM** 4

**Theorem 4.** *The lower bound incurred in the overall generalization error across all $N$ clients of* CORESET-PFEDBAYES *is always higher compared to that of the generalization error in the original full data setup*

$$\left[\frac{1}{N}\sum_{i=1}^{N}\int_{\Theta}d^{2}(\mathcal{P}_{\boldsymbol{\theta},w}^{i},\mathcal{P}^{i})\hat{q}^{i}(\boldsymbol{\theta};\boldsymbol{w})d\boldsymbol{\theta}\right]_{l.b.} \geq \left[\frac{1}{N}\sum_{i=1}^{N}\int_{\Theta}d^{2}(\mathcal{P}_{\boldsymbol{\theta}}^{i},\mathcal{P}^{i})\hat{q}^{i}(\boldsymbol{\theta})d\boldsymbol{\theta}\right]_{l.b.}$$

*Proof.* It is easy to show since from Theorem 3, we know the lower bounds for the individual terms and also since $n > n_k$ holds, hence we can rewrite as follows:

$$\frac{1}{N}\sum_{i=1}^{N}\int_{\Theta}d^{2}(\mathcal{P}_{\boldsymbol{\theta},\boldsymbol{w}}^{i},\mathcal{P}^{i})q^{i}(\hat{\boldsymbol{\theta}},\boldsymbol{w})d\boldsymbol{\theta} - \frac{1}{N}\sum_{i=1}^{N}\int_{\Theta}d^{2}(\mathcal{P}_{\boldsymbol{\theta}}^{i},\mathcal{P}^{i})q^{i}(\hat{\boldsymbol{\theta}})d\boldsymbol{\theta}$$

$$\geq C_3 n_k^{-\frac{2\beta}{2\beta+\Lambda}} - C_3 n^{-\frac{2\beta}{2\beta+\Lambda}}$$

$$\geq 0$$

The implication of this proof states that the overall error incurred due to coreset weighted deviation is always more than that of the original deviation which can be measured approximately in order of $n_k$, the coreset sample size. $\qquad\square$

**Proposition 1.** *The gradient of the first term in Equation 7 i.e.*

$$\nabla_w \mathbb{D}_{KL}(\hat{q}^{i}(\boldsymbol{\theta};\mathbf{w})||\hat{q}^{i}(\boldsymbol{\theta}))$$

*is given by the following expression*

$$\int_{\Theta} \nabla_w \hat{q^i}(\boldsymbol{\theta}; \mathbf{w}) \left[ \log \hat{q^i}(\boldsymbol{\theta}; \mathbf{w}) + 1 - \log \hat{q^i}(\boldsymbol{\theta}) \right] d\boldsymbol{\theta}$$

*where*

$$\nabla_w \hat{q^i}(\boldsymbol{\theta}; \mathbf{w}) \quad = \quad \frac{\hat{q^i}(\boldsymbol{\theta}; \boldsymbol{w})}{\varrho^i(\boldsymbol{\theta_{i,m}}; \boldsymbol{w})} g'_m(\boldsymbol{w}) \quad + \quad g_m(\boldsymbol{w}) \nabla_w \prod_{k \neq m}^{T} \varrho^i(\boldsymbol{\theta_{i,k}}; \boldsymbol{w}) \quad and \quad q^i(\boldsymbol{\theta}; \boldsymbol{w}) \quad = \quad \prod_{m=1}^{T} \varrho^i(\boldsymbol{\theta_{i,m}}; \boldsymbol{w})$$

*Proof.*

$$
\begin{aligned}
& \nabla_w \mathbb{D}_{KL}(\hat{q^i}(\theta; \mathbf{w}) || \hat{q^i}(\theta)) \\
&= \nabla_w \mathrm{E}_{\hat{q^i}(\theta; \mathbf{w})} \left[ \log \hat{q^i}(\theta; \mathbf{w}) - \log \hat{q^i}(\theta) \right] \\
&= \nabla_w \left[ \int_{\Theta} \hat{q^i}(\boldsymbol{\theta}; \mathbf{w}) \log \hat{q^i}(\boldsymbol{\theta}; \mathbf{w}) d\boldsymbol{\theta} - \int_{\Theta} \hat{q^i}(\boldsymbol{\theta}; \mathbf{w}) \log \hat{q^i}(\boldsymbol{\theta}) d\boldsymbol{\theta} \right] \\
&= \left[ \int_{\Theta} \nabla_w \left( \hat{q^i}(\boldsymbol{\theta}; \mathbf{w}) \log \hat{q^i}(\boldsymbol{\theta}; \mathbf{w}) \right) d\boldsymbol{\theta} - \int_{\Theta} \nabla_w \left( \hat{q^i}(\boldsymbol{\theta}; \mathbf{w}) \log \hat{q^i}(\boldsymbol{\theta}) \right) d\boldsymbol{\theta} \right] \\
&= \left[ \int_{\Theta} \left( \log q^i(\hat{\boldsymbol{\theta}}; \mathbf{w}) \nabla_w \hat{q^i}(\boldsymbol{\theta}; \mathbf{w}) + \nabla_w \hat{q^i}(\boldsymbol{\theta}; \mathbf{w}) \right) d\boldsymbol{\theta} - \int_{\Theta} \log \hat{q^i}(\boldsymbol{\theta}) \nabla_w \hat{q^i}(\boldsymbol{\theta}; \mathbf{w}) d\boldsymbol{\theta} \right] \\
&= \int_{\Theta} \nabla_w \hat{q^i}(\boldsymbol{\theta}; \mathbf{w}) \left[ \log \hat{q^i}(\boldsymbol{\theta}; \mathbf{w}) + 1 - \log \hat{q^i}(\boldsymbol{\theta}) \right] d\boldsymbol{\theta}
\end{aligned}
\tag{19}
$$

In order to compute the gradient $\nabla_w \hat{q^i}(\boldsymbol{\theta}; \mathbf{w})$, the following objective can be utilized.

Let $z^*(\boldsymbol{\theta})$ be the optimal variable solution to Equation (5).

$$\nabla_{q^i(\theta)} F_i^w(z^*) \bigg|_{q^i(\hat{\theta}; \mathbf{w})} = 0$$

$$\implies \underbrace{\nabla_{q^i(\theta)} \int_{\Theta} - \log \mathcal{P}_{\theta, w}(\boldsymbol{\mathcal{D}}^i) q^i(\boldsymbol{\theta}) d\boldsymbol{\theta} \bigg|_{q^i(\hat{\theta}; \mathbf{w})}}_{\textbf{First Part}} + \underbrace{\zeta \nabla_{q^i(\boldsymbol{\theta})} \mathbb{D}_{KL}(q^i(\boldsymbol{\theta}) || z^*(\boldsymbol{\theta})) \bigg|_{q^i(\hat{\theta}; \mathbf{w})}}_{\textbf{Second Part}} = 0$$

For the **first part**,

$$
\begin{aligned}
& \nabla_{q^i(\theta)} \int_{\Theta} - \log \mathcal{P}_{\theta, w}(\boldsymbol{\mathcal{D}}^i) q^i(\boldsymbol{\theta}) d\boldsymbol{\theta} \bigg|_{q^i(\hat{\theta}; \mathbf{w})} \\
&= \int_{\Theta} \nabla_{q^i(\theta)} \left[ - \log \mathcal{P}_{\theta, w}(\boldsymbol{\mathcal{D}}^i) q^i(\boldsymbol{\theta}) \right] d\boldsymbol{\theta} \bigg|_{q^i(\hat{\theta}; \mathbf{w})} \\
&= \underbrace{\int_{\Theta} \left[ -q^i(\boldsymbol{\theta}) \nabla_{q^i(\theta)} \log \mathcal{P}_{\theta, w}(\boldsymbol{\mathcal{D}}^i) + \log \mathcal{P}_{\theta, w}(\boldsymbol{\mathcal{D}}^i) \right] d\boldsymbol{\theta} \bigg|_{q^i(\hat{\theta}; \mathbf{w})}}_{\textbf{Modified First part}}
\end{aligned}
\tag{20}
$$

By the assumption that the distribution $q^i(\boldsymbol{\theta})$ satisfies mean-field decomposition i.e.

$$
\begin{aligned}
q^i(\boldsymbol{\theta}) &= \prod_{m=1}^{T} \mathcal{N}(\theta_{i,m}, \sigma_n^2) \\
&= \prod_{m=1}^{T} \varrho^i(\boldsymbol{\theta}_{i,m})
\end{aligned}
\tag{21}
$$

Let us denote $\mathcal{M}_w = \mathcal{P}_{\theta,w}(\boldsymbol{\mathcal{D}}^i)$.

Therefore, we extract out the following portion from (20): $\nabla_{q^i(\theta)} \log \mathcal{P}_{\theta,w}(\boldsymbol{\mathcal{D}}^i)$

$$\nabla_{q^i(\theta)} \log \mathcal{P}_{\theta,w}(\boldsymbol{\mathcal{D}}^i) = \nabla_{q^i(\theta)} \log \mathcal{M}_w \tag{22}$$

We now consider the individual partial differentials here

$$\frac{\partial}{\partial \varrho^i(\boldsymbol{\theta_{i,m}})} \log \mathcal{M}_w = \frac{1}{\mathcal{M}_w} \frac{\partial \mathcal{M}_w}{\partial w} \frac{\partial w}{\partial \varrho^i(\boldsymbol{\theta_{i,m}})} \tag{23}$$

Thus, we can rewrite (20) from the perspective of individual components of $q^i(\boldsymbol{\theta})$ as follows:

$$
\begin{aligned}
&\int_{\Theta} \left[ -q^i(\boldsymbol{\theta}) \frac{1}{\mathcal{M}_w} \frac{\partial \mathcal{M}_w}{\partial w} \frac{\partial w}{\partial \varrho^i(\boldsymbol{\theta_{i,m}})} + \log \mathcal{P}_{\theta,w}(\boldsymbol{\mathcal{D}}^i) \right] d\boldsymbol{\theta} \bigg|_{q^i(\hat{\theta};\mathbf{w})} \\
&= \underbrace{\int_{\Theta} \left[ -q^i(\boldsymbol{\theta}) \frac{1}{\mathcal{P}_{\theta,w}(\boldsymbol{\mathcal{D}}^i)} \frac{\partial \mathcal{P}_{\theta,w}(\boldsymbol{\mathcal{D}}^i)}{\partial w} \frac{\partial w}{\partial \varrho^i(\boldsymbol{\theta_{i,m}})} + \log \mathcal{P}_{\theta,w}(\boldsymbol{\mathcal{D}}^i) \right] d\boldsymbol{\theta}}_{\textbf{Modified First part}} \bigg|_{q^i(\hat{\theta};\mathbf{w})}
\end{aligned} \tag{24}
$$

Now, we can rewrite the **second part** as follows:

$$
\begin{aligned}
&\zeta \nabla_{q^i(\theta)} \mathbb{D}_{KL}(q^i(\boldsymbol{\theta}) \| z^*(\theta)) \bigg|_{q^i(\hat{\boldsymbol{\theta}};\mathbf{w})} \\
&= \zeta \nabla_{q^i(\theta)} \left[ \int_{\Theta} q^i(\boldsymbol{\theta}) \log q^i(\boldsymbol{\theta}) - q^i(\boldsymbol{\theta}) \log(z^*(\boldsymbol{\theta})) d\boldsymbol{\theta} \right] \bigg|_{q^i(\hat{\theta};\mathbf{w})} \\
&= \zeta \int_{\Theta} \nabla_{q^i(\theta)} \left[ q^i(\boldsymbol{\theta}) \log q^i(\boldsymbol{\theta}) - q^i(\boldsymbol{\theta}) \log(z^*(\boldsymbol{\theta})) \right] d\boldsymbol{\theta} \bigg|_{q^i(\hat{\theta};\mathbf{w})} \\
&= \underbrace{\zeta \int_{\Theta} \left( \log q^i(\boldsymbol{\theta}) + 1 - \log(z^*(\boldsymbol{\theta})) \right) d\boldsymbol{\theta}}_{\textbf{Modified Second Part}} \bigg|_{q^i(\hat{\theta};\mathbf{w})}
\end{aligned} \tag{25}
$$

$\square$

Combining both the first and second part we get

$$
\begin{aligned}
&\int_{\Theta} \left[ -q^i(\hat{\boldsymbol{\theta}};\boldsymbol{w}) \frac{1}{\mathcal{P}_{\theta,w}(\boldsymbol{\mathcal{D}}^i)} \frac{\partial \mathcal{P}_{\theta,w}(\boldsymbol{\mathcal{D}}^i)}{\partial w} \frac{\partial w}{\partial \varrho^i(\boldsymbol{\theta_{i,m}})} + \log \mathcal{P}_{\theta,w}(\boldsymbol{\mathcal{D}}^i) \right] d\boldsymbol{\theta} \\
&\qquad\qquad + \zeta \int_{\Theta} \left( \log q^i(\hat{\boldsymbol{\theta}};\boldsymbol{w}) + 1 - \log(z^*(\boldsymbol{\theta})) \right) d\boldsymbol{\theta} = 0 \\
&\implies \zeta \int_{\Theta} \left( \log q^i(\hat{\boldsymbol{\theta}};\boldsymbol{w}) + 1 - \log(z^*(\boldsymbol{\theta})) + \log \mathcal{P}_{\theta,w}(\boldsymbol{\mathcal{D}}^i) \right) d\boldsymbol{\theta} \\
&\qquad\qquad = \int_{\Theta} \left( q^i(\hat{\boldsymbol{\theta}};\boldsymbol{w}) \frac{1}{\mathcal{P}_{\theta,w}(\boldsymbol{\mathcal{D}}^i)} \frac{\partial \mathcal{P}_{\theta,w}(\boldsymbol{\mathcal{D}}^i)}{\partial w} \frac{\partial w}{\partial \varrho^i(\boldsymbol{\theta_{i,m}})} \right) d\boldsymbol{\theta}
\end{aligned} \tag{26}
$$

Let us assume without loss of generality that each of the individual components of the optimal coreset weighted client distribution $q^i(\hat{\boldsymbol{\theta}};\boldsymbol{w})$ can be denoted as some function $g(\boldsymbol{w})$. More, specifically,

$$\varrho^i(\boldsymbol{\theta_{i,j}}; \boldsymbol{w}) = g_j(\boldsymbol{w})$$
$$\nabla_w \varrho^i(\boldsymbol{\theta_{i,j}}; \boldsymbol{w}) = g_j^{'}(\boldsymbol{w})$$

$$(27)$$

Thus we can reuse the above expression to simplify (26)

$$g_m^{'}(\boldsymbol{w}) = \frac{\int_\Theta \left( q^i(\hat{\boldsymbol{\theta}}; \boldsymbol{w}) \frac{1}{\mathcal{P}_{\theta,w}(\boldsymbol{\mathcal{D}^i})} \frac{\partial \mathcal{P}_{\theta,w}(\boldsymbol{\mathcal{D}^i})}{\partial w} \right) d\boldsymbol{\theta}}{\zeta \int_\Theta \left( \log q^i(\hat{\boldsymbol{\theta}}; \boldsymbol{w}) + 1 - \log(z^*(\boldsymbol{\theta})) + \log \mathcal{P}_{\theta,w}(\boldsymbol{\mathcal{D}^i}) \right) d\boldsymbol{\theta}}$$

$$(28)$$

We now go back to utilizing the above derived expression in our main Eq. (19) to replace $\nabla_w q^i(\hat{\boldsymbol{\theta}}; \mathbf{w})$

$$\nabla_w q^i(\hat{\boldsymbol{\theta}}; \boldsymbol{w})$$
$$= \nabla_w \prod_{k=1}^T \varrho^i(\boldsymbol{\theta_{i,k}}; \boldsymbol{w})$$
$$= \nabla_w \prod_{k=1}^T g_k(\boldsymbol{w})$$
$$= \prod_{k \neq m}^T g_k(\boldsymbol{w}) \nabla_w g_m(\boldsymbol{w}) + g_m(\boldsymbol{w}) \nabla_w \prod_{k \neq m}^T g_k(\boldsymbol{w})$$
$$= \prod_{k \neq m}^T g_k(\boldsymbol{w}) g_m^{'}(\boldsymbol{w}) + g_m(\boldsymbol{w}) \nabla_w \prod_{k \neq m}^T g_k(\boldsymbol{w})$$
$$= \frac{q^i(\hat{\boldsymbol{\theta}}; \boldsymbol{w})}{\varrho^i(\boldsymbol{\theta_{i,m}}; \boldsymbol{w})} g_m^{'}(\boldsymbol{w}) + g_m(\boldsymbol{w}) \nabla_w \prod_{k \neq m}^T \varrho^i(\boldsymbol{\theta_{i,k}}; \boldsymbol{w})$$

$$(29)$$

Thus, we now have a closed form solution to computing the gradient of the KL divergence $\mathbb{D}(q^i(\hat{\boldsymbol{\theta}}; \mathbf{w}) || q^i(\hat{\boldsymbol{\theta}}))$ w.r.t the coreset weight parameters $\boldsymbol{w}$.

**Proposition 2.** *The gradient of the second term in Equation 8 w.r.t $\boldsymbol{w}$ i.e.*

$$\nabla_w || P_\theta(\boldsymbol{\mathcal{D}^i}) - P_{\theta,w}(\boldsymbol{\mathcal{D}^i}) ||_{\hat{\pi},2}^2$$

*is given by the following expression*

$$-2\mathcal{P}_{\boldsymbol{\Phi}}^T \left( \mathcal{P} - \mathcal{P}_{\boldsymbol{\Phi}} \boldsymbol{w} \right)$$

*where $\mathcal{P} = \sum_{j=1}^n \hat{g}_j$ and $\mathcal{P}_{\boldsymbol{\Phi}} = [\hat{g}_1, \hat{g}_2, \ldots, \hat{g}_n]$*

*Proof.* First, we reformulate the given expression in terms

$$\left\| P_\theta(\boldsymbol{\mathcal{D}^i}) - P_{\theta,w}(\boldsymbol{\mathcal{D}^i}) \right\|_{\hat{\pi},2}^2$$
$$= \mathrm{E}_{\theta \sim \hat{\pi}}[(P_\theta(\boldsymbol{\mathcal{D}^i}) - P_{\theta,w}(\boldsymbol{\mathcal{D}^i}))^2]$$

We define $g_j = \mathcal{P}_\theta(\boldsymbol{\mathcal{D}}_j^i) - \mathrm{E}_{\theta \sim \hat{\pi}} P_\theta(\boldsymbol{\mathcal{D}}_j^i)$

As a result the equivalent optimization problem becomes minimizing $\left\| \sum_{j=1}^n g_j - \sum_{j=1}^n w_j g_j \right\|_{\hat{\pi},2}^2$

Further, using Monte Carlo approximation, given $S$ samples $\{\theta_j\}_{j=1}^S$, $\theta_j \sim \hat{\pi}$, the $L^2(\hat{\pi})$-norm can be approximated as follows

$$\left\| \sum_{j=1}^n \hat{g}_j - \sum_{j=1}^n w_j \hat{g}_j \right\|_2^2$$

where

$\hat{g}_j = \frac{1}{\sqrt{S}} \left[ \mathcal{P}_{\theta_1}(\boldsymbol{\mathcal{D}}_j^i) - \mathcal{P}(\bar{\boldsymbol{\mathcal{D}}}_j^i), \mathcal{P}_{\theta_2}(\boldsymbol{\mathcal{D}}_j^i) - \mathcal{P}(\bar{\boldsymbol{\mathcal{D}}}_j^i), \ldots, \mathcal{P}_{\theta_S}(\boldsymbol{\mathcal{D}}_j^i) - \mathcal{P}(\bar{\boldsymbol{\mathcal{D}}}_j^i) \right]$ and $\mathcal{P}(\bar{\boldsymbol{\mathcal{D}}}_j^i) = \frac{1}{S} \sum_{k=1}^S \mathcal{P}_{\theta_k}(\boldsymbol{\mathcal{D}}_j^i)$

We can write the above problem in matrix notation as follows

$$f(\boldsymbol{w}) := \|\mathcal{P} - \mathcal{P}_{\boldsymbol{\Phi}} \boldsymbol{w}\|_2^2$$

where $\mathcal{P} = \sum_{j=1}^n \hat{g}_j$ and $\mathcal{P}_{\boldsymbol{\Phi}} = [\hat{g}_1, \hat{g}_2, \ldots, \hat{g}_n]$

Thus we have the gradient w.r.t $\boldsymbol{w}$ as follows:

$$\nabla_w f(\boldsymbol{w}) = -2\mathcal{P}_{\boldsymbol{\Phi}}^T \left( \mathcal{P} - \mathcal{P}_{\boldsymbol{\Phi}} \boldsymbol{w} \right) \tag{30}$$

$\square$

## 10 EXPERIMENTS

All the experiments have been done using the following configuration: Nvidia RTX A4000(16GB) and Apple M2 Pro 10 cores and 16GB memory.

### 10.1 PROPOSAL FOR A MODIFIED OBJECTIVE IN EQUATION 8

$$\{\boldsymbol{w_i}^*\} \triangleq \arg\min_{\boldsymbol{w}} \mathbb{D}_{KL}(\hat{q}^i(\boldsymbol{\theta}, \boldsymbol{w}) \| \hat{q}^i(\boldsymbol{\theta})) + \left\| P_{\boldsymbol{\theta}}(\boldsymbol{\mathcal{D}^i}) - P_{\boldsymbol{\theta}, \boldsymbol{w_i}}(\boldsymbol{\mathcal{D}^i}) \right\|_{\hat{\pi},2}^2 \quad \|\boldsymbol{w_i}\|_0 \leq k \tag{31}$$

We discuss here the utility of our proposed modified client side objective function via an ablation study where we want to gauge the inclusion of the first term in our objective function as just inlcuding the coreset loss.

Through experimental analysis, we find that just including the coreset loss optimization results in early saturation, possibly hinting towards getting stuck in local minima, but however inclusion the KL Divergence loss and forcing the coreset weighted local distribution of the client and the normal local distribution of the client to be similar leads to better stability in the training loss and better convergence.

### 10.2 COMMUNICATION COMPLEXITY ANALYSIS FOR DIFFERENT CORESET SIZES

Here we showcase an analysis for different coreset sample size for different datasets and how it affects on the final accuracy and the total number of communication rounds in the Federated Learning setting. This showcases cost-effectiveness of our approach where by using only a small number of communication rounds our proposed approach is able to attain near-optimal performance as per the table below. In addition Fig: 5 substantiates the cost-effectiveness of our approach.

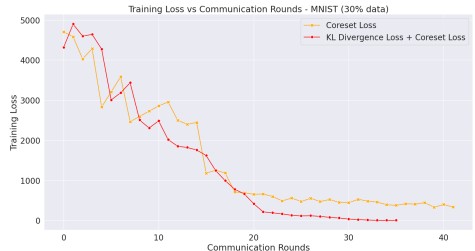 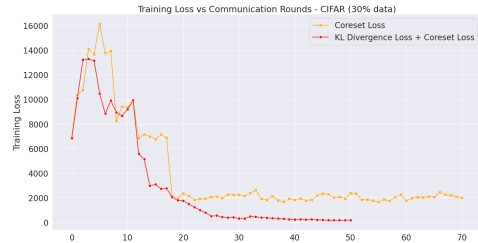

Figure 4: Ablation Study on using KL divergence between two local distribution w.r.t just using coreset weights

Table 3: Comparative results of test accuracies across different coreset sample complexity

| Method (Percentage = sampling fraction) | MNIST | | FashionMNIST | | CIFAR | |
|---|---|---|---|---|---|---|
| | Test Accuracy | Communication Rounds | Test Accuracy | Communication Rounds | Test Accuracy | Communication Rounds |
| PFEDBAYES (Full) | 98.79 | 194 | 93.01 | 215 | 83.46 | 266 |
| RANDOMSUBSET (50%) | 80.2 | 135 | 87.12 | 172 | 48.31 | 183 |
| CORESET-PFEDBAYES (k = 50%) | 92.48 | 98 | 89.55 | 93 | 69.66 | 112 |
| CORESET-PFEDBAYES (k = 30%) | 90.17 | 84 | 88.16 | 72 | 59.12 | 70 |
| CORESET-PFEDBAYES (k = 15%) | 88.75 | 62 | 85.15 | 38 | 55.66 | 32 |
| CORESET-PFEDBAYES (k = 10%) | 85.43 | 32 | 82.64 | 24 | 48.25 | 16 |

(a) We report test accuracies across different sample complexity for datasets like **MNIST, CIFAR, Fashion-MNIST**. **Full indicates training on full dataset and 50% is on using half the data size after randomly sampling 50% of the training set.**

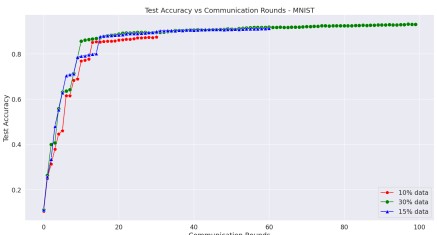 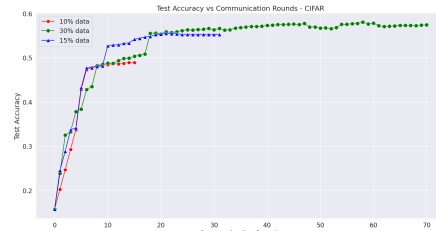

Figure 5: Communication Rounds across Different Sample Size - Convergence analysis

## 10.3 COMPUTING LIKELIHOOD OBJECTIVE USING AIHT

Here, we showcase how we utilised the Accelerated Iterative Hard Thresholding algorithm (A-IHT) for computing the likelihood.

## 10.4 MEDICAL DATASET EXPERIMENT DETAILS

Owing to the rise of Federated Learning based approaches in the medical setting due to privacy-preserving features, we chose to perform our experiments on 3 medical datasets in addition to our main experiments.

For our Federated Learning setup, we considered the setting where we have only 2 clients and one global server.

For each of the 3 datasets in the medical dataset setting, we consider each client has X-ray images of symptomatic type **A**/ type **B** and Normal images . We perform a classification task at each client.

## 10.5 BASLINE COMPARISONS: DIVERSITY BASED SUBMODULAR OPTIMIZATION FUNCTIONS

For our second set of experiments, we chose different diversity based submodular optimization functions, specifically the following functions whose definition have been provided here

**Definition 4.** *Log-determinant Function is a diversity-based submodular function. It is non-monotone in nature. Let $\mathbf{L}$ denote a positive semidefinite kernel matrix and $\mathbf{L_S}$ denote the subset of rows and columns indexed by set $\mathbf{S}$. Log-determinant function $f$ is specified as:*

$$f(\mathbf{S}) = logdet(\mathbf{L_S}) \tag{32}$$

The log-det function models diversity and is closely related to a determinantal point process.

**Definition 5.** *Disparity Sum Function characterizes diversity by considering the sum of distances between every pair of points in a subset $\mathbf{S}$. For any two points $i, j \in \mathbf{S}$, let $d_{ij}$ denote the distance between them.*

$$f(\mathbf{S}) = \sum_{i,j \in \mathbf{S}} d_{ij} \tag{33}$$

*The aim is to select a subset $\mathbf{S}$ such that $f(\mathbf{S})$ is maximized. Disparity sum is not a submodular function.*

**Definition 6.** *Disparity Min Function characterizes diversity by considering the minimum distance between any two non-similar points in a subset $\mathbf{S}$.*

$$f(\mathbf{S}) = \min_{i,j \in \mathbf{S}, i \neq j} d_{ij} \tag{34}$$

*The aim is to select a subset $\mathbf{S}$ such that $f(\mathbf{S})$ is maximized. Disparity min is not a submodular function.*

For the above experiments we utilise the *Submodlib library* [3] for our implementation Kaushal et al. (2022).

## 10.6 EXPERIMENT CONFIGURATION

### 10.6.1 MNIST EXPERIMENT CONFIGURATION

For both CORESET-PFEDBAYES and corresponding baseline PFEDBAYES, we use a fully connected DNN model with 3 layers [784,100,10] on MNIST dataset.

**Learning rate hyperparameters**: As per Zhang et al. (2022b)'s proposal i.e. PFEDBAYES the learning rates for personalized (client model) and global model ($\eta_1, \eta_2$) are set to 0.001 since these choices result in the best setting for PFEDBAYES. To compare against the stable best hyperparameters of PFEDBAYES, we also fix the same for our proposal CORESET-PFEDBAYES.

**Personalization Hyperparameter**: The $\zeta$ parameter adjusts the degree of personalization in the case of clients. Again for a fair comparison against our baseline PFEDBAYES, we fix the $\zeta$ parameter for our proposal CORESET-PFEDBAYES to the best setting given by the baseline. In Zhang et al. (2022b) the authors tune $\zeta \in \{0.5, 1, 5, 10, 20\}$ and find that $\zeta = 10$ results in the best setting. We, therefore, fix the personalization parameter $\zeta = 10$.

### 10.6.2 MEDICAL DATASETS EXPERIMENT CONFIGURATION

We discuss here the detailed configuration and models used for our further experiments.

Here we specifically consider the setting where we only have *2* clients and a single global server. Each of the 2 clients are assigned with data from only 2 classes along with a shared class for classification purpose.

For example, client 1 has class $A$ and *Normal* (shared class) images while client 2 has class $B$ and the remaining *Normal* images.

**COVID-19 Radiography Database**: Client 1 has COVID-19 x-ray images while client 2 has lung opacity x-ray images. Normal X-ray images are shared across both clients. Fig. 6 depicts the dataset distribution. For random subset selection, we randomly choose $\lambda = 0.1$ fraction of samples on the client side. For diversity-based subset selection, we first convert each of the 299×299 images into a [512×1] vector embeddings using a ResNet architecture. Diversity functions are then applied to

---

[3] Submodlib decile library

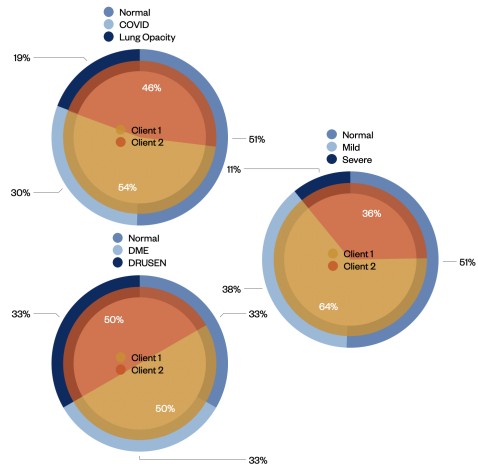

Figure 6: Data distribution for Medical Datasets

these embeddings to retrieve a final subset of diverse and representative embeddings. Eventually, we decode back to the original space using the chosen representative indices.

**APTOS 2019 Blindness Detection**: Unlike the COVID-19 radiography dataset, the APTOS dataset has 3 RGB channels and a higher resolution. We rescale the dimension of images to 299x299 for maintaining uniformity across all datasets. The same model configuration is followed as in the COVID-19 radiography dataset.

**OCTMNIST**: The OCTMnist dataset is a large dataset with single-channel images of a higher resolution. We have resized the images to $299 \times 299$ resolution for our experiments. The Normal class has above 50,000 train images itself, with the other two classes having close to 10,000 train images. Due to this class imbalance, we have randomly selected 8,000 images from each class for our experiments. Post which we again use a ResNet architecture to reduce the feature dimensions, which we then feed into the CORESET-PFEDBAYES pipeline.

**Baseline : Independent Learning** In this scenario, each of the 2 clients solve the classification problem independently without any involvement of a server as opposed to federated learning. Thus there is no sharing of model weights to a common server as compared to the federated setting.

**Baseline : Independent Learning on other client's test data** In this scenario, similar to the independent learning setup, we report the metrics for a particular client not only on its own test data but also on the other client's test data by training on the individual client's own training data.

For all the experiments for the medical dataset analysis across all the baselines, we report the class-wise accuracy in Table 2.

**Definition 7.** *Submodular Functions are set functions which exhibit diminishing returns. Let* $\mathbf{V}$ *denote the ground-set of $n$ data points $\{x_1, x_2, \ldots x_n\}$ where $x_i \in \mathbb{R}^d$. More formally,* $\mathbf{V} = \{x_i\}_{i=1}^n$. *Let $\mathbf{A} \subseteq \mathbf{B}$ where $\mathbf{A}, \mathbf{B} \subset \mathbf{V}$ and $v \in \mathbf{V}$. A submodular function $f : 2^V \mapsto \mathbb{R}$ satisfies the diminishing returns property as follows:*

$$f(\mathbf{A} \cup v) - f(\mathbf{A}) \geq f(\mathbf{B} \cup v) - f(\mathbf{B}) \tag{35}$$

## 10.7 ALGORITHM FOR ACCELERATED IHT

We first present the accelerated IHT algorithm as proposed in Zhang et al. (2021) in Algorithm 10.7.

---

**Algorithm 2** Accelerated IHT (A-IHT) for Bayesian Coreset Optimization

---

**Input Objective** $f(w) = \|y - \Phi w\|_2^2$; sparsity $k$

1: $t = 0, z_0 = 0, w_0 = 0$

2: repeat

3: $\quad \mathcal{Z} = \mathrm{supp}\,(z_t)$

4: $\quad \mathcal{S} = \mathrm{supp}\,\left(\Pi_{\mathcal{N}_k \setminus \mathcal{Z}}\,(\nabla f\,(z_t))\right) \cup \mathcal{Z}$ where $|\mathcal{S}| \le 3k$

5: $\quad \widetilde{\nabla}_t = \nabla f\,(z_t)|_{\mathcal{S}}$

6: $\quad \mu_t = \arg\min_\mu f\left(z_t - \mu\widetilde{\nabla}_t\right) = \dfrac{\left\|\bar{\nabla}_t\right\|_2^2}{2\left\|\Phi\bar{\nabla}_t\right\|_2^2}$

7: $\quad w_{t+1} = \Pi_{\mathcal{C}_k \cap \mathbb{R}_+^n}\,(z_t - \mu_t \nabla f\,(z_t))$

8: $\quad \tau_{t+1} = \arg\min_\tau f\,(w_{t+1} + \tau\,(w_{t+1} - w_t))$

$\quad = \dfrac{\langle y - \Phi w_{t+1}, \Phi(w_{t+1} - w_t)\rangle}{2\|\Phi(w_{t+1} - w_t)\|_2^2}$

9: $\quad z_{t+1} = w_{t+1} + \tau_{t+1}\,(w_{t+1} - w_t)$

10: $\quad t = t + 1$

11: until Stop criteria met

12: return $w_t$

---

The algorithm **Accelerated IHT** above is proposed by Zhang et al. (2021). We share a high level view of the algorithm include some of the important features.

**Step Size Selection** The authors propose that given the quadratic objective of the coreset optimization, they perform exact line search to obtain the best step size per iteration. $\dfrac{\left\|\bar{\nabla}_t\right\|_2^2}{2\left\|\Phi\bar{\nabla}_t\right\|_2^2}$

**Momentum** The authors propose adaptive momentum acceleration as is evident from line 8 of the pseudocode. At the end during the next update, Nesterov Accelerated Gradient is applied as shown in line 9.

## 11 CODE

We share our code on GitHub at Link

