# OpenReview forum: "Bayesian Coreset Optimization for Personalized Federated Learning"
_ICLR.cc/2024/Conference — ICLR 2024 poster_

### Official Review · Reviewer_FxK1 · 2023-10-31

**Soundness:** 3 good
**Presentation:** 3 good
**Contribution:** 3 good
**Rating:** 8
**Confidence:** 2

**Summary:**

The work incorporated granular-level bayesian coresets optimization in Federated Learning. The proposed approach gave minimax convergence rate and showed good performance in empirical studies.

**Strengths:**

1. The idea of incorporating coreset optimization in FL is new and well-motivated.
2. Solid theoretical results are given.
3. Some optimistic empirical studies are presented.

**Weaknesses:**

1. The major weakness is the lack of convergence comparison in the empirical part. One of the major concerns in FL is the communication cost. Thus the number of iteration rounds is crucial in FL. The reviewer suggests not only including the comparison of the final accuracy under (maybe different levels, not only 50%) of sample complexity, but also including the convergence speed, i.e., the communication cost comparison.

2. How expensive it is to calculate the coreset samples/weights? Is there any empirical runtime results?

3. How is \hat{\pi} defined in Eq. (3) and (4)?

4. Some typo: first sentence in section 3.2 is incomplete. Different places for \hat notation in q^i(\theta, w), on q, or q^i, or q^i(theta, w).

**Questions:**

See weakness.

---

> ### Author Response · Authors · 2023-11-18
> **Official Response 1/2 to the Reviewer FxK1**
>
> We appreciate the reviewer's time and effort in reviewing our paper on employing Bayesian coresets in federated learning. Their insightful feedback is invaluable, and we are thankful for the constructive comments provided.
>
> We greatly appreciate the reviewer's positive evaluation of our work, particularly their recognition of the well-motivated approach in incorporating coreset optimization into federated learning. The reviewer's acknowledgment of the our proposed solid theoretical results and the optimism reflected in our empirical studies is truly motivating.
>
> **Weakness**
>
>     The major weakness is the lack of convergence comparison in the empirical part. One of the major concerns in FL is the communication cost. Thus the number of iteration rounds is crucial in FL. The reviewer suggests not only including the comparison of the final accuracy under (maybe different levels, not only 50%) of sample complexity, but also including the convergence speed, i.e., the communication cost comparison.
>
>
>  We appreciate the reviewer's attention to the critical aspect of convergence comparison and its relevance to the communication cost in Federated Learning (FL). While we may have already provided in our theoretical results the convergence rate achieved in the case of our method which is in order of logarithmic bounds of the coreset size $k$ and thereby much faster than full data training,  we also get better convergence as compared to random sampling (as indicated in Fig 3). In addition to these, we update on our empirical results to include different subset sample sizes along with their communication cost (total no. of communication rounds) as follows.
>
>
> In Appendix Section 10.2 we are including new results to show case the total number of communication rounds (where under each setting where we are considering different coreset subsample sizes k=50,30,15,10) which is indicative of the convergence of our proposed method. Empirical results shows that with significant lesser coreset subsample sizes we achieve faster convergence with less number of communication rounds , but however with lesser accuracy than that if we used a larger coreset size or full data training which is as expected.
>
>
>
> **Table: Comparative results of test accuracies across different coreset sample complexity**
>
> | Method (Percentage = sampling fraction) | MNIST                       |                             | FashionMNIST                    |                             | CIFAR                       |                             |
> |-----------------------------------------|-------------------------------|-----------------------------|-----------------------------------|-----------------------------|-----------------------------|-----------------------------|
> |                                         | Test Accuracy                | Communication Rounds       | Test Accuracy                    | Communication Rounds       | Test Accuracy                | Communication Rounds       |
> |-----------------------------------------|-------------------------------|-----------------------------|-----------------------------------|-----------------------------|-----------------------------|-----------------------------|
> | $pFedBayes$ (Full)                      | 98.79                         | 194                         | 93.01                             | 215                         | 83.46                       | 266                         |
> | $pFedBayes$ with random sampling (50%)                      | 80.2                          | 135                         | 87.12                             | 172                         | 48.31                       | 183                         |
> | $Our Method$ (k = 50%)                  | 92.48                         | 98                          | 89.55                             | 93                          | 69.66                       | 112                         |
> | $Our Method$ (k = 30%)                  | 90.17                         | 84                          | 88.16                             | 72                          | 59.12                       | 70                          |
> | $Our Method$ (k = 15%)                  | 88.75                         | 62                          | 85.15                             | 38                          | 55.66                       | 32                          |
> | $Our Method$ (k = 10%)                  | 85.43                         | 32                          | 82.64                             | 24                          | 48.25                       | 16                          |

---

> ### Author Response · Authors · 2023-11-18
> **Official Response 2/2 to the Reviewer FxK1**
>
> How expensive it is to calculate the coreset samples/weights? Is there any empirical runtime results?
>
> Our approach to computing coreset samples/weights is exceptionally cost-effective, thanks to the utilization of accelerated iterative hard thresholding in the coreset—an incredibly efficient method for rapidly determining coreset weights. Although we haven't conducted explicit runtime experiments for coreset sample weights, we've assessed the overhead cost in terms of computation, revealing an average of only 0.001 seconds per round with a maximum of 10 iterations for coreset computation (which is ideal due to early convergance in coreset weights)  per round. Over 200 rounds, this results in an additional overhead cost of just 0.2 seconds compared to random sampling.
>
> We look forward to suggestions regarding details to any additional specific runtime experiments that the reviewer would like us to try out.
>
>      How is $\hat{\pi}$ defined in Eq. (3) and (4)?
>
> We have defined $\hat{\pi}$ as the weighting distribution that has the same support under an embedding Hilbert space as the true posterior $\pi$, for which we consider an L2 norm as the distance metric between the client's overall data likelihood and its corresponding coreset weighted data likelihood.
>
>
>      Some typo: first sentence in section 3.2 is incomplete. Different places for \hat notation in $q^i(\theta, w)$, on $q$, or $q^i$, or $q^i(theta, w)$.
>
> Thank you for your careful inspection. We have identified this and similar places if any and have fixed the same in our revised version. Additionally regarding the notation for hat notation, $q^i(\theta, w)$ we have chosen to keep it consistent to $\hat{q^i}$. We have identified the sections where there is this notation inconsistency like in Section 4 and Section 6 of the main paper and have fixed them in our final revision. There are currently some portions in the supplementary which we are converting to a consistent notation for the hat operator and which will updated in our upcoming rebuttal revision.
>
>
> ---------------------------------------------------------------
>
> We hope that the rebuttal clarifies the questions raised by the reviewer. We would be happy to discuss any further questions about the work, and would appreciate an appropriate increase in the score if the reviewer’s concerns are adequately addressed.

---

> > ### Comment · Reviewer_FxK1 · 2023-11-20
> > **Response**
> >
> > The reviewer has read the feedback given by the authors. The reviewer believe the added experimental results has addressed most of my concerns, thus raising the score.

---

> ### Author Response · Authors · 2023-11-21
> **Response to Reviewer FxK1**
>
> We are deeply grateful to the reviewer for the positive support for our paper and the increment in score. We would appreciate the reviewer's support for acceptance of the paper.

---

### Official Review · Reviewer_QB83 · 2023-10-31

**Soundness:** 3 good
**Presentation:** 2 fair
**Contribution:** 2 fair
**Rating:** 5
**Confidence:** 3

**Summary:**

This paper introduces an optimization framework for personalized federated learning by incorporating Bayesian coresets into the model proposed in [1]. The author want to ensure that the accuracy performance does not deteriorate when applying coresets. To achieve this, they have made modifications to the common coreset objective. Furthermore, they provide proof of the convergence rate of generalization error using their approach and evaluate the effectiveness of their method on a range of datasets.

[1] Xu Zhang, Yinchuan Li, Wenpeng Li, Kaiyang Guo, and Yunfeng Shao. Personalized federated learning via variational bayesian inference.

**Strengths:**

- The integration of Bayesian coresets with federated learning is innovative.
- In the context of personalized federated learning, this work presents new ideas and considerations for defining the objective in coreset computation, which differs from the commonly used coreset definition.

**Weaknesses:**

- The paper's content is a bit bloated, and the use of notations can be messy. For instance, sections 3.2 and 4 could be condensed to make them more concise. Additionally, there is potential to simplify the formulaic aspect.
- It would be beneficial if the author could emphasize their novel contribution, distinguishing it from the techniques previously proposed by others. Currently, these ideas seem to be mixed within the intricate details of the interpretations.
- The overall architecture, as well as certain smaller techniques and theoretical analysis methods, seem to be largely derived from previous work.
- The contribution on the coreset construction is limited. Although the authors introduce a new coreset objective, they do not provide sufficient NEW optimization techniques for the new objective. I could only identify some techniques borrowed from previous work.
- In my opinion, the primary contribution of this paper is the modified objective (eq. 9) tailored to personalized federated learning. However, the advantages of this modified objective are not adequately elucidated in the current presentation.

some minor problems

- In section 3, there is a confusion of n and N. For example, n in Fig 1 should be N.
- In section 3.2 , it should be $ g_j = \mathcal{P}_\theta(\mathcal{D}_j^i) = E_{\theta\sim \hat{\pi}} P_\theta(\mathcal{D}_j^i) $.
- The subscript of the bold variable should not be bolded if it is a scalar.
- many other typos, e.g. missing equation references and confusing sentence like “For the first term in Equation 1, the authors we use a minibatch stochastic gradient descent …”

**Questions:**

- What is the benefits to apply coreset in the personalized federated learning? I think one of the most important is that it can reduce the communication complexity. It would be valuable to investigate and quantify the extent to which the coreset approach reduces communication complexity in the specific optimization task addressed in this work. This can be done theoretically, by providing a complexity formula, and practically, by presenting numerical results from experiments that show the reduction in communication complexity achieved.
- the intuition behind the new objective in eq. 9 is not very persuasive. If you could compute a coreset with a sufficiently small loss as defined in eq. 3, it is unecessary to add the term representing the “distance” between $\hat{q}^i(\theta, w)$ and $\hat{q}^i(\theta)$ since $\hat{q}^i(\theta, w)$ and $\hat{q}^i(\theta)$ will lead to closed losses; On the other hand, if you couldn’t make it under the constraint $\| w \|_0 \leq k$, which means there is no such small coreset with ideal error, the coreset method could not work well. It would be beneficial to clarify the merits of the new objective, such as its robustness or any other advantages it offers. Experiments that demonstrate the effectiveness of the new objective would greatly strengthen the argument.
- Does the modifications of eq. 6 consist of the following two parts: i) use the weighted likelihood. ii) replace prior distribution with global distritution. I am not sure for that.
- is there any strategy for choosing the value of k in practice?

---

> ### Author Response · Authors · 2023-11-18
> **Official Response 1/4 to the Reviewer QB83**
>
> We extend our gratitude to the reviewer for dedicating time and effort to assess our paper on the utilization of Bayesian coresets in federated learning.
>
> We appreciate the reviewer's recognition of the innovation in integrating Bayesian coresets framework within the federated learning setting in our paper. The reviewer's positive assessment of this aspect is encouraging, and we are pleased that they find our approach to be innovative.
> We are equally appreciative of the reviewer's acknowledgement for our novel research proposal and new ideas in this space which we believe will be significantly beneficial to the community.
>
> Now, we go forward to clarify some of the questions raised by the reviewer.
>
> **Weaknesses**
>
>     The paper's content is a bit bloated, and the use of notations can be messy. For instance, sections 3.2 and 4 could be condensed to make them more concise. Additionally, there is potential to simplify the formulaic aspect.
>
> We thank the reviewer for their valuable feedback and insights on the paper's content and notations. In response, we have streamlined and condensed sections 3.2 and 4 to enhance conciseness without compromising the clarity of our presentation, which has been reflected in our updated rebuttal version. We look forward to the reviewer's suggestions in understanding which other specific portions can be further simplified to improve upon our current draft.
>
>     It would be beneficial if the author could emphasize their novel contribution, distinguishing it from the techniques previously proposed by others. Currently, these ideas seem to be mixed within the intricate details of the interpretations.
>
> Our **primary objective** in this proposal is to **enhance the efficiency in Federated Learning training via utilizing a fraction of the original data at each client** while maintaining **near-optimal accuracy**, along with **minimally possible communication cost**. Our proposed theoretical findings showcase that **model trained on a client's coreset sample size weighted data show faster convergence rate** than in the case when trained fully on the entire training data.
>
> Both Bayesian Coreset Optimization and Federated Learning Optimization problems are **two different orthogonal and independent problems**. Considering their both distinct challenges, we aim to infuse the two optimization framework in one single unified cohesive framework such that the coreset optimization problem can **strategically pick data points for each client** in a **model dependent manner**, i.e. without facing too much penalty in the model training under Federated Learning setup. As of now, to the best of our knowledge, this specific problem **remains largely unexplored**. Remarkably, there is a significant gap in the existing research landscape concerning subset selection or coreset-based strategies within the context of Federated Learning, with a conspicuous absence of theoretical frameworks in this domain.
>
> Crucially, **our work is unique in addressing the absence of research on subset selection or coreset-based strategies within Federated Learning setup** and theoretical contributions along with empirical results in terms of near optimal performance under data scarcity setting accompany our contributions.
>
> Lastly we also **showcase its applicability in medical domain scenario** via medical datasets where **data scarcity remains a huge problem** and how our proposal can alleviate the same.
>
>      The overall architecture, as well as certain smaller techniques and theoretical analysis methods, seem to be largely derived from previous work.
>
> We thank for the reviewer's feedback in this aspect. However, we respectfully assert a differing viewpoint. While building upon prior research to establish a robust foundation, our **contribution is distinctive in its innovative synthesis and adaptation of Bayesian Coresets and Personalized Federated Learning frameworks**. The fusion of these domains creates a novel architecture of its own as it can be seen in Algorithm 1 with unique optimization objectives, thereby further leading to new theoretical formulations and findings, as evidenced in our Theoretical Analysis and extensive novel formulations in the Supplementary (Section 9 Lemma 1, Proposition 1, Proposition 3, Proposition 4. ).  We further refine and extend these objectives to enhance performance and efficiency. We firmly believe that the synergy of existing knowledge with our novel contributions significantly strengthens the overall impact and applicability of our work. We aim to further clarify certain aspects of the above thread with specific instances as we go forward.

---

> ### Author Response · Authors · 2023-11-18
> **Official Response 2/4 to the Reviewer QB83**
>
> The contribution on the coreset construction is limited. Although the authors introduce a new coreset objective, they do not provide sufficient NEW optimization techniques for the new objective. I could only identify some techniques borrowed from previous work.
>
> We appreciate the reviewer's feedback. It is essential to emphasize that our primary contribution lies not in introducing new optimization techniques for the Bayesian Coreset construction but instead, our innovation centers around **proposing a comprehensive algorithm and corresponding optimization strategies for a common unifying framework** that governs two distinct optimization problems at hand. This approach leads to a **model-dependent coreset sampling technique** (in this case the model being trained under a personalized federated learning setup) i.e. along with the model training for federated learning, the coreset weights are also learnt dynamically for each client, which is a novel aspect of our contribution. While drawing inspiration from previous work is a common practice, our work's strength lies in the integration and adaptation of these techniques to address the challenges posed by our unique framework. In particular for bayesian coreset optimization problem, we **propose a novel objective loss in Eq 9/8** on top of which we apply Accelerated Iterative Hard Thresholding. It's noteworthy that while AIHT is an efficient optimization strategy in its own right, our emphasis is not on its intrinsic properties. Rather our **proposed optimization objective stands out for its considerable complexity** and to address the same, we **employ optimization strategies tailored to its nuances**, such as the use of AIHT. Detailed insights into this choice are provided in the Supplementary section proofs, affirming our confidence in the appropriateness of this approach.
>
>
>       In my opinion, the primary contribution of this paper is the modified objective (eq. 9) tailored to personalized federated learning. However, the advantages of this modified objective are not adequately elucidated in the current presentation.
>
>
> We appreciate your recognition of the main contribution being the modified objective function (Eq. 9) tailored for personalized federated learning. Your feedback on the need for a more thorough elucidation of the advantages is duly noted. In Section 3.2, we delve into the rationale and motivation behind our proposed approach, offering a more comprehensive understanding of the modified objective's benefits. Additionally, we have conducted new experiments and updated our explanations in response to Question 2 under **Questions**, specifically addressing the advantages of the modified objective. This enhancement aims to provide a clearer and more detailed exposition of the strengths associated with our proposed contribution.

---

> ### Author Response · Authors · 2023-11-18
> **Official Response 3/4 to the Reviewer QB83**
>
> **Questions**
>
>       What is the benefits to apply coreset in the personalized federated learning? I think one of the most important is that it can reduce the communication complexity. It would be valuable to investigate and quantify the extent to which the coreset approach reduces communication complexity in the specific optimization task addressed in this work. This can be done theoretically, by providing a complexity formula, and practically, by presenting numerical results from experiments that show the reduction in communication complexity achieved.
>
> We value the reviewer's input on comprehending the computational complexity and the effectiveness of our approach in this context. Consequently, we perform an analysis of communication complexity to assess the total number of communication rounds required for full convergence across various coreset subsample sizes. We present our results as follows (Additionally we have included these results under Supplementary section 10.2)
>
>
> | Method (Percentage = sampling fraction) | MNIST                       |                             | FashionMNIST                    |                             | CIFAR                       |                             |
> |-----------------------------------------|-------------------------------|-----------------------------|-----------------------------------|-----------------------------|-----------------------------|-----------------------------|
> |                                         | Test Accuracy                | Communication Rounds       | Test Accuracy                    | Communication Rounds       | Test Accuracy                | Communication Rounds       |
> |-----------------------------------------|-------------------------------|-----------------------------|-----------------------------------|-----------------------------|-----------------------------|-----------------------------|
> | $pFedBayes$ (Full)                      | 98.79                         | 194                         | 93.01                             | 215                         | 83.46                       | 266                         |
> | $pFedBayes$ with random sampling (50%)                      | 80.2                          | 135                         | 87.12                             | 172                         | 48.31                       | 183                         |
> | $Our Method$ (k = 50%)                  | 92.48                         | 98                          | 89.55                             | 93                          | 69.66                       | 112                         |
> | $Our Method$ (k = 30%)                  | 90.17                         | 84                          | 88.16                             | 72                          | 59.12                       | 70                          |
> | $Our Method$ (k = 15%)                  | 88.75                         | 62                          | 85.15                             | 38                          | 55.66                       | 32                          |
> | $Our Method$ (k = 10%)                  | 85.43                         | 32                          | 82.64                             | 24                          | 48.25                       | 16                          |

---

> > ### Author Response · Authors · 2023-11-18
> > **Official Response 4/4 to the Reviewer QB83**
> >
> > **Question 2** the intuition behind the new objective in eq. 9 is not very persuasive. If you could compute a coreset with a sufficiently small loss as defined in eq. 3, it is unecessary to add the term representing the “distance” between $\hat{q}^{i}(\theta, w)$
> > and $\hat{q}^{i}(\theta)$ since $\hat{q}^{i}(\theta, w)$ and $\hat{q}^{i}(\theta)$ will lead to closed losses; On the other hand, if you couldn’t make it under the constraint, which means there is no such small coreset with ideal error, the coreset method could not work well. It would be beneficial to clarify the merits of the new objective, such as its robustness or any other advantages it offers. Experiments that demonstrate the effectiveness of the new objective would greatly strengthen the argument.
> >
> > **Response**
> > We appreciate your thoughtful engagement with our work and would like to address the concerns regarding the intuition of the proposed objective in Equation 9. Your point about the potential redundancy of the first term in Eq 9. i.e. the KL divergence term, given that we are already considering the second term i.e. the coreset loss term, is a valid claim. However, we would like to offer a different viewpoint regarding the same. In addition to our discussions in Section 3.2, we provide further clarifications as to our motivation behind including the first term in Eq 9, along with corresponding empirical observations to justify the same. Note the second term captures the deviation of the data likelihood over the entire client's data w.r.t that of the coreset weighted client data. When the second term loss is small under specific sparsity constraints, it leads to the conclusion that a suitable $\boldsymbol{w}$ has been found such that coreset weighted client's data likelihood mimics to a good extent to that of the original client's data likelihood. The above of choice of $\boldsymbol{w}$ can surely be then utilised to achieve a corresponding solution to the minimization problem of client side objective: Eq 6. (which is say $\hat{q^{i}}(\theta, w)$. \textbf{However, this optimal $\hat{q^{i}}(\theta, w)$ distribution (i.e. local distribution of the coreset weighted client) can be any arbitrary distribution acting as a minima to Eq 6. (as there might be many such local minimas for the same), which further leads to a different optimal variational solution for the global problem in Eq 2. (say $\boldsymbol{z^*(\boldsymbol{\theta},w)}$ ) , which can be quite different than the optimal variational solution for the global setting in the case when the entire client's data is considered (say $\boldsymbol{z^*(\boldsymbol{\theta})}$), thereby not leading to an optimal solution at the end.
> >
> > This was also initially observed empirically when we just tried out with the coreset loss as our main objective function. In Supplementary section 10.1 we do a study to check why the KL divergence term (more specifically forcing the local client's distribution under a coreset setting mimic that of the local client's distribution under a full data training setting) is beneficial.
> >
> > Based on our experimental analysis, we observe that exclusively incorporating the coreset loss optimization may lead to early saturation, suggesting a potential susceptibility to getting trapped in local minima. However, introducing the KL Divergence loss and enforcing similarity between the coreset-weighted local distribution of the client and the normal local distribution yields improved stability in the training loss and enhances convergence.
> >
> > Corresponding plots : https://pasteboard.co/3VGSZG0B4ab1.png (MNIST)
> >                                   https://pasteboard.co/4pXBqEIhy5Ik.png (CIFAR)
> >
> >
> > **Question 3** Does the modifications of eq. 6 consist of the following two parts: i) use the weighted likelihood. ii) replace prior distribution with global distritution. I am not sure for that.
> >
> > **Response** : Yes correct, the modifications of eq. 6 consist of the above parts
> >
> > **Question 4** is there any strategy for choosing the value of k in practice?
> > **Response** : The consideration depends on the specific application setting we are looking at in terms of resource/data constraints and the level of sacrifice in terms of accuracy acceptable for the given context. While we cannot directly quantify the accuracy in relation to k, for a particular k (since a lot of other factors weigh in like class-wise distribution and others), the goal here is to achieve optimal accuracy that approaches the original accuracy with considerable gaps. The determination is influenced by the trade-offs we are willing to make.
> >
> > **minor typos**: We have identified the same and currently fixing the same for our next rebuttal version. We will update shortly.
> >
> > We hope that the rebuttal clarifies the questions raised by the reviewer. We would be happy to discuss any further questions about the work, and would appreciate an appropriate increase in the score if the reviewer’s concerns are adequately addressed.

---

> > > ### Author Response · Authors · 2023-11-21
> > > **Thank you for the helpful feedbacks and suggestions.**
> > >
> > > Dear Reviewer QB83, We want to thank you for your encouraging comments and helpful suggestions to make our proposal stronger. We have addressed your questions in the response and incorporated your suggestions in the revised manuscript. Please let us know if you have any further questions, and we would be happy to address them within our allowed period. If the concerns are addressed adequately, we would appreciate an appropriate increase in the score.

---

> ### Comment · Reviewer_QB83 · 2023-11-21
>
> Thank you for your detailed response. I believe that the coreset optimization offers advantages in terms of the communication complexity. And the additional experiments indicate that the coreset has better performance than random sampling.  However, the comparision to the full data is not very competitive since the sampling size is of the same magnitude as the full data, which is also mentioned by other reviewers. The authors point out that the coreset method leads to smaller communication rounds. Therefore, i would like to see the performances of the full data and coresets of varying sizes in the case where the number of communication rounds is limited to different thresholds. It would be valuable to assess the advantages of coresets in scenarios where only a limited number of communication rounds are allowed. If such advantages are demonstrated, I would be inclined to raise my rating accordingly.

---

> ### Author Response · Authors · 2023-11-22
> **Thank you Reviewer QB83 for your response and helpful suggestions**
>
> Dear Reviewer QB83,
>
> We appreciate your helpful feedback and suggestion towards the importance of testing across multiple communication rounds threshold. We are therefore reporting new results on Test accuracy coresponding to the settings where only a limited number of communication rounds are now allowed (limited communication rounds settings 15,25,40 and 70 across all 3 datasets and for different sample subset size threshold).
>
> This is in addition to the previous results as requested https://openreview.net/forum?id=uz7d2N2zul&noteId=dkzGM5rRfF where we are not considering any limitation on the number of communication rounds.
>
> | Method (Percentage = sampling fraction) |   MNIST                    |                             |                             |                             |   FashionMNIST               |                             |                             |                             |   CIFAR                     |                             |                             |                             |
> |-----------------------------------------|-----------------------------|-----------------------------|-----------------------------|-----------------------------|-----------------------------------|-----------------------------|-----------------------------|-----------------------------|-----------------------------|-----------------------------|-----------------------------|-----------------------------|
> |                                         | Communication Rounds (at 15)| Communication Rounds (at 25)| Communication Rounds (at 40)| Communication Rounds (at 70)| Communication Rounds (at 15)| Communication Rounds (at 25)| Communication Rounds (at 40)| Communication Rounds (at 70)| Communication Rounds (at 15)| Communication Rounds (at 25)| Communication Rounds (at 40)| Communication Rounds (at 70)|
> |-----------------------------------------|-----------------------------|-----------------------------|-----------------------------|-----------------------------|-----------------------------------|-----------------------------|-----------------------------|-----------------------------|-----------------------------|-----------------------------|-----------------------------|-----------------------------|
> | $pFedBayes$ (Full)                      | 92.44                           | 93.17                           | 94.02                           | 94.77                           | 88.91                        | 89.20                           | 89.27                           | 90.33                           | 53.94                           | 59.54                           | 68.13                           | 72.77                           |
> | $Our Method$ (k = 50%)                  | 87.13                          | 91.62                      | 91.47                          | 91.94                         | 80.26                          | 83.57                      | 86.22                          | 89.13                          | 54.66                          | 59.11                      | 61.04                          | 64.58                          |
> | $Our Method$ (k = 30%)                  | 86.45                          | 89.12                      | 89.36                          | 90.15                          | 80.01                          | 84.55                     | 85.81                          | 88.28                        | 52.43                          | 55.95                      | 57.38                          | 59.12                          |
> | $Our Method$ (k = 15%)                  | 78.14                          | 85.17                      | 87.81                          | 88.73                          | 79.72                          | 83.41                     | 85.15                          | 85.14                          | 48.91                          | 54.96                      | 55.63                          | 55.81                          |
> | $Our Method$ (k = 10%)                  | 73.68                          | 84.26                      | 85.43	                          | 85.51                          | 75.13                          | 82.64                      | 82.69                          | 82.71                          | 48.25                          | 48.26                      | 48.22                           | 48.29                          |
>
> We hope that the rebuttal clarifies the questions raised in this aspect.

---

> > ### Comment · Reviewer_QB83 · 2023-11-22
> >
> > Thank you for your response. I have decided to increase my score for the following reasons:
> >
> > i) The modifications made to the presentation.
> >
> > ii) I believe that the coreset method shows potential advantages, particularly in terms of communication complexity, in the personalized federated learning.
> >
> > iii) The experiments conducted in the paper demonstrate that the proposed coreset method outperforms random sampling.
> >
> > However, I have not increased my score significantly because the experiments indicate that there is no particular need to apply the coreset method in this Bayesian-PFL framework. The size of the coreset is comparable to that of the full data, and the performance is worse than that of the full data when the numbers of communication rounds are the same.

---

> ### Author Response · Authors · 2023-11-23
> **Thank you Reviewer QB83 for Helpful Feedback**
>
> Dear Reviewer QB83,
>
> We deeply appreciate the increase in score and yours positive acknowledgement on the concerns raised earlier in terms of the motivation for our framework and experimental analysis in efficiency and outperformance over random sampling.
>
> We would like to point our here as well, that when we considered around 50% of sample size, that leads to around half the chunk of the training data that is lost. We have further decremented it to 10% data (i.e. only considering 10% of the original training size). In any case when we train on only a few samples as opposed to the entire training set we would expect less accuracy.  10% , 25% and 40% and 50% data are significantly smaller chunks of our original training data and hence it is imperative that we have a low performance as compared to full data training. In all these cases the sample size is very much less in the order than that of training set (even in the case of 50%).
>
> Our main experimental contribution is to show case the utility of coreset optimization in the case of Bayesian-PFL framework over random sampling of training data points or even applying popular submodular function based subset selection strategies where we get very less significant drop in accuracy (achieve near optimal accuracy as compared to the baselines).

---

### Official Review · Reviewer_YU5h · 2023-11-02

**Soundness:** 3 good
**Presentation:** 2 fair
**Contribution:** 3 good
**Rating:** 6
**Confidence:** 3

**Summary:**

The paper describes a method to use Bayesian coresets for each individual client in a federated learning setting. Bayesian coreset can be used as proxy for full data at each individual client to estimate client-side distribution. The authors describe objective functions to incorporate the Bayesian coresets with federated learning setting. The authors give an algorithm and also give theoretical guarantees for the generalization error and its convergence. The authors support their theoretical claims with empirical results comapring their proposed approach with a number of baselines.

**Strengths:**

1. The paper is, for the most part, well written. There is not much work in terms of coresets for federated learning and as such the paper will be of interest to the community.
2. The authors have compared their method with a variety of baselines consisting of both - federated learning algorithms and also sampling strategies that incorporate diversity.  Their method performs well in most of the cases.
3. The algorithm is backed with theoretical guarantees. I did not check the proofs, but the statements appear sound.

**Weaknesses:**

1. I am not sure what is the challenge in incorporating the Bayesian coreset framework in federated learning setting. It would be better to explain clearly why this is a significant contribution. Both the algorithm and proof techniques appear to be heavily inspired from Zhang 2022b. The only modification seems to be use of Bayesian coresets.

2. There are minor grammatical errors. Please do a grammar check.

**Questions:**

1. Why the prior $\pi$ in equation 1 is replaced by $\mathbf{z}$ in eq.6 - the modified client-side objective. Please clarify.

2. The subsample size is 50%. Is it not quite large? Does it give significant computational time benefits when compared with full data? Other than figure 3, there are no experiments mentioning computational efficiency.

3. Not a question but a suggestion. Algorithm 1 is not easy to follow for anyone unfamiliar with existing work or similar algorithms. How exactly is the coreset getting constructed? It would be good to give a high-level description of the same.

Overall, the paper appears sound and I would be happy to raise my score once the doubts are cleared.

---

> ### Author Response · Authors · 2023-11-18
> **Official Response 1/3 to the Reviewer YU5h**
>
> We sincerely appreciate the time and effort that the reviewer has invested in reviewing our paper on utilizing Bayesian coresets in a federated learning setting. The reviewer's insightful feedback is invaluable to us, and we are grateful for the constructive comments that they have provided.
>
> Firstly, we are pleased to hear that the reviewer has found the paper to be well-written and acknowledges its potential significance to the community given the lack of such work in this space.
> Secondly we are thankful to the reviewer for acknowledging the thoroughness of our comparative analysis and theoretically motivated framework and that they found our method's performance commendable across a variety of scenarios, especially when benchmarked against both federated learning algorithms and diverse sampling strategies. The reviewer's positive assessment encourages us and reaffirms the effectiveness of our proposed approach.
>
>
> **Weaknesses**:
>
>      I am not sure what is the challenge in incorporating the Bayesian coreset framework in federated learning setting. It would be better to explain clearly why this is a significant contribution. Both the algorithm and proof techniques appear to be heavily inspired from Zhang 2022b. The only modification seems to be use of Bayesian coresets.
>
> One of the **core problems that we try to address** here is how to **achieve efficiency in training in a Federated Learning setup** **using only a fraction of the original data at each client** yet achieving **near optimal performance in terms of accuracy** (theoretical results on our logarithm bounded convergence rate using coreset indicates better convergence than full training while empirical results across a diverse set of baselines including diversity based subset selection techniques show promising results in terms of accuracy). Both the Bayesian Coreset optimization framework and Personalized Federated Learning framework  which in itself involves a  bilevel optimization problem are two separate entirely different orthogonal problems.
>
> Our goal in this proposal is thus to **bridge the two towards a common optimization framework** whereby we can utilise a Bayesian coreset framework for strategic selection of data points at each client level and at the same time we do not need to pay for a bigger penalty while solving the Federated Learning optimization problem. Both these problems represent independent optimization challenges that need to be addressed concurrently. The difficulty arises when striving to learn the coreset in an optimized manner that minimizes the overall penalty incurred while training within the Federated Learning setting. Hence our work focuses on seamlessly **interleaving these two optimization frameworks** into a **single, cohesive framework**, introducing **novel optimization formulations as evident in Eq 8 and 7** (updated equation numbers due to manuscript revision), along with corresponding strong theoretical guarantees. Importantly, our *contribution addresses a gap in the existing literature*. To the best of our knowledge, this specific problem has not been thoroughly explored. There is a notable absence of research on subset selection or coreset-based strategies within a Federated Learning context, let alone the formulation of theoretical frameworks in this domain.
>
>
> Additionally, as an application setting we also showcase the usefulness and applicability of our method on **real-world medical datasets where in many cases there is a severe dearth of medical data**.
> To address the challenges in infusing the two optimization frameworks together and propose new findings on top of it, we draw upon some of the proof techniques from papers like Polson
> & Roˇckov ́a (2018), and others, one of them being Zhang 2022b, particularly due to the similarities observed in Eq. 1. We utilize certain existing lemmas and theorems, on top of which we provide novel theoretical formulations and new results as seen in theorem 1 and theorem 2, and their accompanied proofs in Supplementary (Section 9 Lemma 1, Proposition 1, Proposition 3, Proposition 4. ) which are our primary contributions.
>
> [1] Nicholas G Polson and Veronika Roˇckov ́a. Posterior concentration for sparse deep learning.
>
>      There are minor grammatical errors. Please do a grammar check.
>
> We thank the reviewer for their careful inspection. We have made a thorough proofread pass and have identified the specific areas and have fixed them in our recent revision.

---

> ### Author Response · Authors · 2023-11-18
> **Official Response 2/3 to the Reviewer YU5h**
>
> **Questions**
>
>
>     Why the prior in equation 1 is replaced by *z* in eq.6 - the modified client-side objective. Please clarify.
>
> Thank you for your inquiry regarding the substitution of the prior in Equation 1 with 'z' in Equation 6. In Equation 1, we initially introduce the concept of a prior distribution when formalizing the client-side objective. For the sake of clarity and ease of understanding, we denote this prior probability distribution as $\pi$ as it represents a more nuanced notation to indicate probability and also as it encompasses additional terms such as $\pi(\theta|D)$. In Equation 6, we streamline and simplify the representation while retaining the essence of the prior distribution in a more concise form using *z*. We have also adequately defined after the equation to resolve any confusion for the same.
>
>     The subsample size is 50%. Is it not quite large? Does it give significant computational time benefits when compared with full data? Other than figure 3, there are no experiments mentioning computational efficiency.
>
> We do agree that taking subsample size = 50% can be considered somewhat large, but at the same time dropping too many data points may also result in degradation in performance (accuracy) irrespective of what sampling strategies we use. Hence we thought 50% to be a respectful threshold where we can measure the effectiveness of our proposed approach for maintaining near-optimal performance as compared to other baselines. Keeping in consideration the importance of studying across different thresholds we have included some of the baseline comparisons on less percentage data as follows.
>
> In Appendix Section 10.2 we are including new results to show case the total number of communication rounds (per setting where we consider different coreset subsample sizes k=50,30,15,10) which is indicative of the convergence of our proposed method. Empirical results shows that with significant lesser coreset subsample sizes we achieve faster convergence with less number of communication rounds but with lesser accuracy than that if we used a larger coreset size or full data training.
> **Table: Comparative results of test accuracies across different coreset sample complexity**
>
> | Method (Percentage = sampling fraction) | MNIST                       |                             | FashionMNIST                    |                             | CIFAR                       |                             |
> |-----------------------------------------|-------------------------------|-----------------------------|-----------------------------------|-----------------------------|-----------------------------|-----------------------------|
> |                                         | Test Accuracy                | Communication Rounds       | Test Accuracy                    | Communication Rounds       | Test Accuracy                | Communication Rounds       |
> |-----------------------------------------|-------------------------------|-----------------------------|-----------------------------------|-----------------------------|-----------------------------|-----------------------------|
> | $pFedBayes$ (Full)                      | 98.79                         | 194                         | 93.01                             | 215                         | 83.46                       | 266                         |
> | $pFedBayes$ with random sampling (50%)                      | 80.2                          | 135                         | 87.12                             | 172                         | 48.31                       | 183                         |
> | $Our Method$ (k = 50%)                  | 92.48                         | 98                          | 89.55                             | 93                          | 69.66                       | 112                         |
> | $Our Method$ (k = 30%)                  | 90.17                         | 84                          | 88.16                             | 72                          | 59.12                       | 70                          |
> | $Our Method$ (k = 15%)                  | 88.75                         | 62                          | 85.15                             | 38                          | 55.66                       | 32                          |
> | $Our Method$ (k = 10%)                  | 85.43                         | 32                          | 82.64                             | 24                          | 48.25                       | 16                          |

---

> ### Author Response · Authors · 2023-11-18
> **Official Response 3/3 to the Reviewer YU5h**
>
> Not a question but a suggestion. Algorithm 1 is not easy to follow for anyone unfamiliar with existing work or similar algorithms. How exactly is the coreset getting constructed? It would be good to give a high-level description of the same.
>
> In Algorithm 1, our approach involves evaluating the client side objective function both the full data-based likelihood and the coreset weighted likelihood for each client. Subsequently, we calculate the objective function using Equation 8 ( now changed Eq number due to revised manuscript). To compute the coreset samples, we employ the highly effective Accelerated IHT algorithm atop this objective function. Supplementary Section 10.5 provides comprehensive details on the Accelerated IHT algorithm for a deeper understanding of its application in our methodology.
>
> ----------------------------------------------------------------------------------------------------------
>
> We hope that the rebuttal clarifies the questions raised by the reviewer. We would be happy to discuss any further questions about the work, and would appreciate an appropriate increase in the score if the reviewer’s concerns are adequately addressed.

---

> > ### Comment · Reviewer_YU5h · 2023-11-21
> > **Thank you for the response**
> >
> > I have read the reviews and response from the author and am slightly more positively inclined towards accepting the paper.

---

> > > ### Author Response · Authors · 2023-11-21
> > > **Response to Reviewer Yu5h**
> > >
> > > We are deeply thankful towards the reviewer's positive support for our paper and the confidence they provided. We look forward to the possibility of an increased score that aligns with the reviewer's positive assessment.

---

> > > > ### Author Response · Authors · 2023-11-22
> > > > **Thank you Reviewer Yu5h for the helpful feedback and suggestions**
> > > >
> > > > Dear Reviewer Yu5h, Thank you again for the helpful suggestions and the important feedbacks which we have incorporated in the paper and for your reassurance given your positive inclination towards accepting the paper.
> > > >
> > > > We would appreciate an appropriate increase in the score so as to get support on the paper acceptance from your end.

---

> > > > > ### Author Response · Authors · 2023-11-23
> > > > > **Response to Reviewer Yu5h (Addressing questions/ concerns and Rating)**
> > > > >
> > > > > Dear Reviewer Yu5h,
> > > > >
> > > > > Once again, Thank you for the careful feedback towards our proposal.
> > > > >
> > > > > Since there are only a few hours left till author response, we would highly appreciate if you could let us know regarding the increase in rating since we have resolved all the previous concerns raised and you have mentioned to increase the score if so.
> > > > >
> > > > > Also if any further queries we would be happy to address during this time period.

---

### Author Response · Authors · 2023-11-21
**Rebuttal Revision on Minor Edits/Typos**

We deeply appreciate the reviewers' helpful feedback and meticulous inspection, and we've incorporated their suggestions into our revision.

- Fixed existing grammatical mistakes (as suggested by R1)
- Fixed mismatch notation for n and N in Figure 1 ( as suggested by R2)
- Fixed \hat operator notation inconsistencies across main draft and supplementary (as suggested by R3)
- Added new experiments under supplementary as suggested by Reviewers
- Fixed any Eq. references as suggested by Reviewers
- Fixed bold subscript issues which are scalar variables (as suggested by Reviewer 2)

---

### Author Response · Authors · 2023-11-23
**Thank you Reviewers**

We extend our sincere gratitude for the valuable feedback the reviewers provided during the review process. The reviewer's insights and suggestions have significantly enhanced our rebuttal revision.

We are particularly grateful for the positive encouragement the reviewers expressed toward our proposal and the acknowledgment of the experimental and theoretical analysis emphasizes the effort we've invested in crafting a comprehensive and innovative approach.

The reviewer's recognition of our proposal as well-motivated and novel in integrating Bayesian Coreset Optimization within a Personalized Federated Learning framework is immensely appreciated. We are excited about the prospect of contributing to a new paradigm in this domain, combining two optimization frameworks in a novel manner.

The acknowledgment of our proposed objectives, supported by theoretical results and guarantees on convergence, reinforces our confidence in the potential for achieving near-optimal performance in terms of accuracy in an efficient manner as compared to random sampling baselines which includes other state of the art FL algorithms as well as popular submodularity based subset selection strategies.

Once again, we express our gratitude for the reviewer's constructive feedback, which has undoubtedly strengthened our work.

---

### Meta-Review · Area_Chair_14Ac · 2023-12-09

**Metareview:**

The paper presents CORESET-PFEDBAYES, integrating Bayesian coreset optimization into personalized federated learning. This approach aims at efficient training on client-specific data points, theoretically minimizing generalization and approximation errors. Empirical results indicate improved model accuracy on several datasets. The authors addressed some reviewer concerns in the discussions, particularly regarding methodology and additional empirical results. Yet, there are remaining concerns about the theoretical novelty and practical efficacy (particularly relative to full dataset training), which would require the author to throughly address should the paper be accepted.

**Justification For Why Not Higher Score:**

The practical advantages of the proposed method, in terms of computational efficiency and communication costs, are not compelling when benchmarked against full dataset training. Another minor issue is that, the proposed approach, while innovative in concept, heavily relies on existing theoretical frameworks (this does not imply a negative evaluation of the paper, but does not help boosting the theoretical significance). These factors, coupled with concerns over clarity and presentation raised by reviewers, suggest that while the work is a step forward in federated learning, it can be further improved in a future revision.

**Justification For Why Not Lower Score:**

The paper offers a sound theoretical basis and empirical evidence supporting its claims. The integration of Bayesian coresets into personalized federated learning is a novel direction in this research area, and the paper demonstrates some level of empirical improvement over baseline methods. The authors' engagement in the review discussions, particularly in addressing questions about methodology and providing additional empirical evidence, further supports the paper's standing.

---

### Decision · Program_Chairs · 2024-01-16

Accept (poster)